# Rethinking Graph Super-Resolution: Dual Frameworks for Topological Fidelity

## Abstract

Graph super-resolution is an underexplored yet highly relevant research direction that circumvents the need for costly and time-consuming data collection, preparation, and storage. This makes it especially desirable for resource-constrained fields such as the medical domain. Existing work on graph super-resolution leverages graph neural networks (GNNs) and achieves impressive results. However, we note two major limitations in the current model design: (1) It violates the underlying graph structure when increasing the number of nodes, and (2) it relies heavily on node representation learning, which has limited capacity to accurately model edges. To address these limitations, we propose two novel frameworks: (1) Bi-SR, which performs structure-aware node super-resolution, and (2) DEFEND, which focuses on edge representation learning for enhanced edge modeling. We supplement our work with rigorous theoretical analysis and conduct extensive experiments on simulated and real-world datasets covering diverse graph topologies and low-to-high resolution relationships. The results demonstrate substantial improvements across all experiments, highlighting the potential of both frameworks for graph super-resolution tasks.

## 1 Introduction

High-resolution (HR) datasets are crucial for accurate analysis and information processing. However, acquiring HR datasets is resource-intensive, necessitating the development of super-resolution techniques to enhance the quality of easily accessible low-resolution (LR) datasets. Consequently, super-resolution has been extensively studied for images, and numerous traditional and deep learning methods have been developed to tackle this challenge (Dong et al., 2015; Greenspan, 2009; Lu et al., 2019; Wang et al., 2022). While images form a significant class of datasets, many real-world problems are naturally and effectively represented using relational structures such as graphs. Examples include traffic flows, molecular structures, brain connectivity, and social interactions.

Despite the ubiquity of graph-structured datasets, graph super-resolution remains underexplored. Unlike images, the LR and HR graphs lack a hierarchical or local relationship, which forms a critical limitation in model design. Considering that the basic building blocks of an image are pixels, locality of image super-resolution allows the use of the de-convolution operator (Zeiler et al., 2010) to easily increase the number of pixels or the size of the image. Similarly for graphs, nodes form the basic building blocks, and an unpooling operation (Gao & Ji, 2019) has been defined to increase the number of nodes or the overall graph size. However, this operation is overly simplistic, highly localized, and requires the connectivity information of the HR graph as input, making it unsuitable for graph super-resolution. Moreover, the lack of hierarchy leads to a significant distributional shift between LR and HR graphs, further amplifying the complexity of graph super-resolution tasks.

Although challenging, graph super-resolution is a highly relevant task, especially in the field of network neuroscience. The connectivity strength between different regions of the brain can be encoded as a brain graph, commonly known as a connectome. Various studies show that HR connectomes lead to better neural fingerprinting and behavior prediction (Tian et al., 2021; Hayasaka & Laurienti, 2010; Zalesky et al., 2010; Finn et al., 2015; Cengiz & Rekik, 2019). However, brain graphs are typically dense and computationally intensive to collect, process, and store, even small graphs, sometimes requiring gigabytes per individual (Tian et al., 2021). Therefore, deep learning methods for lightweight, on-the-fly calculation of HR brain graphs are advantageous.

Recently, graph neural networks (GNNs) have emerged as de facto deep learning methods to process graph-structured datasets (Zhou et al., 2020; Bronstein et al., 2017; Wang et al., 2021) and have naturally been extended for graph super-resolution. Even though they achieve impressive results, we note two major limitations: (1) The operation used to increase the number of nodes relies on a simple linear algebraic technique that maps LR feature dimensions to HR nodes, ignoring the graph structure of the problem. (2) Since most GNNs perform node representation learning, the models use computationally intensive message-passing layers to learn a single node feature capable of encoding all incident edges. These layers are not only unscalable for larger graphs but, by predominately operating in the node space, offer limited capacity to learn graph topology. Combined, these form a significant research gap in graph super-resolution.

The topological limitation presents a serious bottleneck, particularly for applications in network neuroscience. Numerous studies (Pereira et al., 2015; 2016; Khazaee et al., 2015; Nigro et al., 2022; Mijalkov et al., 2017) have shown that brain graph topology plays a central role in correctly identifying the onset and existence of various neurodegenerative disorders, including the two most frequent ones: Alzheimer's disease (AD) and Parkinson's disease (PD). Notably, Pereira et al. (2016) observes that different stages of AD show decreasing path length and mean clustering compared to the control group. Similarly, Pereira et al. (2015) analyzes topological measures like clustering coefficient, characteristic path length, and small-worldness from 3T MRI data, observing aberrant values are for early PD patients. Finally, Nigro et al. (2022) shows a correlation between the loss of hubs in certain brain regions and the emergence of more hubs in others for frontotemporal dementia.

**Our Contributions:** Motivated by above findings, we propose two new frameworks to tackle both limitations of existing graph super-resolution methods: **Bi-SR** (**Bi**partite Graph for **S**uper-**R**esolution) and **DEFEND** (**D**ual Graphs for **E**dge **Fe**ature Learning a**n**d **D**etection). Bi-SR super-resolves nodes through bipartite connections between LR and HR nodes in a way that respects the underlying graph structure of the problem. DEFEND employs a dual graph formulation that maps edges to dual nodes and directly performs edge representation learning using simple GNN layers. We provide comprehensive theoretical analysis to justify the design of our frameworks and substantiate claims regarding their utility. We also conduct extensive experimentation across different graph topologies and LR-HR relationships to showcase performance improvements from both frameworks.

**Related Work:** While the research on graph super-resolution is scarce, few foundational works have made notable contributions. Isallari & Rekik (2021) introduced a graph U-Net architecture (Gao & Ji, 2019), incorporating a hierarchical structure and a graph Laplacian operator for up-sampling LR brain graphs (Tanaka, 2018). Pala et al. (2021) accelerated model training by using representation template graphs at both low and high resolutions as priors. Mhiri et al. (2021) employed NNConv layers (Simonovsky & Komodakis, 2017) for global graph alignment and a graph-GAN model (Wang et al., 2018) to generate HR connectomes. However, this state-of-the-art model struggles with dense brain graphs, often resulting in out-of-memory (OOM) errors due to the computational complexity of NNConv layers. Finally, Monti et al. (2018) uses a similar dual graph formulation to learn attention weights in GAT layers but differs from our work as we are leveraging the dual graphs for direct edge feature learning in graph super-resolution.

## 2 Preliminaries

### 2.1 Graph Data Structure

Graphs are relational data structures defined by $\mathcal{G} = (\mathcal{V}, \mathcal{E}, \mathbf{A}, \mathbf{X})$, where $\mathcal{V}$ is the set of nodes, $\mathcal{E}$ represents edges as ordered/unordered pairs $(v_i, v_j)$ s.t. $v_i, v_j \in \mathcal{V}$, $\mathbf{A} \in \mathbb{R}^{n \times n}$ is the adjacency matrix capturing edge weights, and $\mathbf{X} \in \mathbb{R}^{n \times d}$ is the node feature matrix with $n = |\mathcal{V}|$ nodes and $d$-dimensional features. In this work, we focus on simple undirected graphs, where $\mathbf{A}$ is symmetric with $\mathbf{A}ij = \mathbf{A}ji = e_{ij}$ denoting the relationship strength between nodes $v_i$ and $v_j$. For notational convenience, we use $\mathbf{x}_i$ to represent feature vector for node $v_i$.

### 2.2 Graph Super-Resolution

Let $\mathcal{G}_l = (\mathcal{V}_l, \mathcal{E}_l, \mathbf{A}_l, \mathbf{X}_l)$ and $\mathcal{G}_h = (\mathcal{V}_h, \mathcal{E}_h, \mathbf{A}_h, \mathbf{X}_h)$ represent the LR and HR graphs, respectively, with $\mathcal{G}_l$ obtained from $\mathcal{G}_h$ via a degradation operator $Deg$ with parameter $\delta$ as $\mathcal{G}_l = Deg(\mathcal{G}_h; \delta)$.

The goal of graph super-resolution is to approximate the HR graph $\hat{\mathcal{G}}_h = (\hat{\mathcal{V}}_h, \hat{\mathcal{E}}_h, \hat{\mathbf{A}}_h, \hat{\mathbf{X}}_h)$ using a super-resolution operator $\mathcal{S}$ with parameters $\theta$ as:

$$\hat{\mathcal{G}}_h = \mathcal{S}(\mathcal{G}_l; \theta) \tag{1}$$

Optimal parameters $\hat{\theta}$ are learned by minimizing some loss function $\mathcal{L}(\hat{\mathcal{G}}_h, \mathcal{G}_h)$. Since there can be multiple mappings $\mathcal{S} : \mathcal{G}_l \mapsto \hat{\mathcal{G}}_h$ minimizing $\mathcal{L}$, having prior knowledge of $Deg$ is beneficial. However, unlike image super-resolution, where $Deg$ operates locally and convolutional layers can be used, graph super-resolution lacks locality, requiring a more complex $\mathcal{S}$.

### 2.3 Message Passing Graph Neural Networks

Graph Neural Networks (GNNs) (Zhou et al., 2020) are designed for graph-structured data, which, unlike images, are irregular with no fixed node order or neighborhood size. Message Passing Neural Networks (MPNNs) (Gilmer et al., 2017), a common GNN subclass, handle this irregularity by iteratively updating node features based on messages from neighbors. Theoretically, let $\mathcal{G} = (\mathcal{V}, \mathcal{E}, \mathbf{A}, \mathbf{X}^0)$ be the input graph, where $\mathbf{A}$ is the adjacency matrix and $\mathbf{X}^0$ the initial node features. An $L$ layer MPNN updates node features at layer $l$ via two key operations:

1. **Neighborhood aggregation**: $\mathbf{z}_i^l = \beta^l \mathbf{x}_i^{l-1} + (1 - \beta^l) \sum_{j \in \mathcal{N}_i} \alpha_{ij}^l \mathbf{x}_j^{l-1}$, where $\mathbf{x}_i^{l-1}$ and $\mathbf{x}_j^{l-1}$ are node features from the previous layer, $\mathcal{N}_i$ is the set of neighboring nodes for node $i$, $\alpha_{ij}^l$ is the importance of node $j$ for node $i$ and typically depends on $\mathbf{x}_i^{l-1}$ and $\mathbf{x}_j^{l-1}$, and $\beta^l$ balances the node's own features against the aggregated neighborhood message.

2. **Node feature update**: $\mathbf{x}_i^l = f_n(\mathbf{z}_i^l)$, where $f_n$ is a universal function approximator.

This process, succinctly represented as: $\mathbf{X} = GNN_{mp}(\mathbf{X}^0, \mathbf{A})$ is agnostic to the number of nodes, neighborhood size, and node ordering, making $GNN_{mp}$ equivariant to node permutations i.e $\mathbf{X}^P = GNN_{mp}(\mathbf{P}\mathbf{X}^0, \mathbf{P}\mathbf{A}\mathbf{P}^T) = \mathbf{P}GNN_{mp}(\mathbf{X}^0, \mathbf{A}) = \mathbf{P}\mathbf{X}$ for permutation matrix $\mathbf{P}$.

### 2.4 Problem Statement 1: Structure-aware Super-resolution

However, this method disrupts structural integrity by arbitrarily mapping LR feature dimensions to HR nodes, akin to mapping image channels to pixels, thus losing data structure and failing to support permutation-invariant applications. To address this, our work investigates structure-aware alternatives that preserve the graph's underlying structure, replacing the linear algebraic operator $Transpose(GNN_{n_h}(\mathbf{X}_l, \mathbf{A}_l))$, aiming for a more faithful pixel-to-pixel-like mapping in graph processing.

While numerous methods exist for learning node feature matrix $\hat{\mathbf{X}}$ and structure $\hat{\mathbf{A}}$ for graphs with fixed number of nodes, graph super-resolution requires an operator that expands the number of nodes from $n_l$ to $n_h$. Existing work use a linear algebraic trick to predict $\hat{\mathbf{X}}_h$ from $\mathbf{X}_l \in \mathbb{R}^{n_l \times d}$. First, a GNN maps $\mathbf{X}_l$ to $n_h$-dimensional feature space as $\hat{\mathbf{X}}_l = GNN_{n_h}(\mathbf{X}_l, \mathbf{A}_l)$, where $GNN_{n_h} : \mathbb{R}^{n_l \times d} \mapsto \mathbb{R}^{n_l \times n_h}$. Then, its transpose initializes HR node feature matrix as $\hat{\mathbf{X}}_h = \hat{\mathbf{X}}_l^T$, where $\hat{\mathbf{X}}_h \in \mathbb{R}^{n_h \times n_l}$, and could be used with any downstream task like predicting $\hat{\mathbf{A}}_h$ or HR node classification.

Although effective, this method loses structural integrity by mapping LR feature dimensions to HR nodes—analogous to arbitrarily mapping image channels to pixels, which violets data structure. It is also incompatible with downstream applications requiring node permutation invariance. Therefore, we explore graph structure-aware alternatives to replace this linear algebraic operator $Transpose(GNN_{n_h}(\mathbf{X}_l, \mathbf{A}_l))$ s.t. its akin to mapping pixels to pixels in image processing.

### 2.5 Problem Statement 2: Edge Representation Learning

Traditional GNNs focus on node representation learning, encoding node feature matrix $\hat{\mathbf{X}} = GNN(\mathbf{X}, \mathbf{A})$ into a feature space suitable for given task. For example, the learned feature space for node clustering encodes similar nodes together while maximizes the distance between dissimilar nodes. Edge weights in $\hat{\mathbf{A}}$ are often derived by taking a dot product $\hat{\mathbf{A}} = \mathbf{X} \cdot \mathbf{X}^T$, under the

assumption that a powerful enough GNN would capture all pairwise interactions with its neighbors in a single node representation. However, even complex GNNs may fail to achieve this in practice.

Therefore, we hypothesize and later prove that an alternative approach based on edge representation learning shows higher modeling capacity, allowing the use of simpler GNNs to achieve similar or better performance. Formally, we aim to expand $\hat{\mathbf{A}} = DotProduct(GNN(\mathbf{X}, \mathbf{A}))$ to $\hat{\mathbf{A}} = f_e(\mathbf{X}, \mathbf{A})$, where $f_e$ is an arbitrary composition of other operators, including GNNs, edge space transformations, etc., and has higher edge learning capacity than the dot product operator.

# 3 PROPOSED BI-SR FRAMEWORK

To tackle the structural limitation of the linear algebraic method from section 2.4, we introduce a bipartite graph formulation which creates direct connections between low and high resolution nodes:

---

**Bipartite Graph Formulation**

Let $\mathcal{G}_l = (\mathcal{V}_l, \mathcal{E}_l, \mathbf{A}_l)$ and $\mathcal{G}_h = (\mathcal{V}_h, \mathcal{E}_h, \mathbf{A}_h)$ be low and high resolution graphs. Let $n_l = |\mathcal{V}_l|$ and $n_h = |\mathcal{V}_h|$. We create a complete bipartite graph $\mathcal{G}_b = (\mathcal{V}_b, \mathcal{E}_b, \mathbf{A}_b)$ between nodes in the low resolution and high resolution s.t.:

1. $\mathcal{V}_b = \mathcal{V}_l \cup \mathcal{V}_h$ with $|\mathcal{V}_b| = n_l + n_h$

2. $\mathcal{E}_b = \{(w, v) | w \in \mathcal{V}_l, v \in \mathcal{V}_h\}$ with $|\mathcal{E}_b| = n_l \times n_h$

3. The adjacency matrix $\mathbf{A}_b$ is given as the block matrix:

$$\mathbf{A}_b = \begin{pmatrix} \mathbf{0} & \mathbf{B} \\ \mathbf{B}^T & \mathbf{0} \end{pmatrix} \tag{2}$$

where, $\mathbf{0}$ refers to zero matrices and $\mathbf{B} \in \mathbb{R}^{n_l \times n_h}$ s.t. all entries are 1.

---

Below are two ways to use this bipartite structure to learn HR node features $\hat{\mathbf{X}}_h$ from LR node features $\hat{\mathbf{X}}_l$, where $\hat{\mathbf{X}}_l = GNN(\mathbf{X}_l, \mathbf{A}_l)$ projects LR node features to a suitable space:

**Linear Combination:** This method flexibly initializes each HR node as a linear combination of LR node features: $\hat{\mathbf{X}}_h = \mathbf{W}^b \hat{\mathbf{X}}_l$. Here, $\mathbf{W}^b \in \mathbb{R}^{n_h \times n_l}$ are learnable parameters with $\mathbf{W}^b_{pq}$ indicating the contribution of LR node $q$ to HR node $p$. This effectively learns unique values for each edge in $\mathcal{E}_b$. Moreover, the node feature dimensions remain unchanged; if $\hat{\mathbf{X}}_l \in \mathbb{R}^{n_l \times d_l}$, then $\hat{\mathbf{X}}_h \in \mathbb{R}^{n_h \times d_l}$.

**Message Passing:** While linear combination offers flexibility, message passing intuitively leverages graph structure. However, it requires initial node features for all nodes in $\mathcal{V}_b$. These are easily initialized for LR nodes as $\hat{\mathbf{X}}_l \in \mathbb{R}^{n_l \times d_l}$ but no prior information is available for HR nodes. Let these unknown HR features be $\mathbf{X}_h^0 \in \mathbb{R}^{n_h \times d_l}$ and let's analyze the message passing update to devise an initialization strategy: $\hat{\mathbf{x}}_{ph} = f(\mathbf{x}_{ph}^0 + \sum_{q \in \mathcal{N}_p} \alpha_{pq} \mathbf{x}_{ql})$, where $\mathbf{x}_{ph}^0$ and $\mathbf{x}_{ql}$ are the $p$th and $q$th row of $\mathbf{X}_h^0$ and $\mathbf{X}_l$, respectively. Since $\mathcal{G}_b$ is a complete bipartite graph, all HR nodes share the same neighborhood $\mathcal{N}_p$, making $\mathbf{x}_{ph}^0$ the sole differentiating term for $\alpha_{pq}$ and the message passing update. Therefore, any initialization of $\mathbf{X}_h^0$ must ensure unique embeddings for HR nodes to avoid feature collapse and performance degradation on downstream tasks requiring individual node identification.

To this end, we randomly initialize $\mathbf{X}_h^0$ with values sampled from $\mathcal{U}(0, 1)$. In high dimensional feature space, by the law of large numbers (Hsu & Robbins, 1947), concentration of measure phenomena (Ledoux, 2001), and the Johnson-Lindenstrauss Lemma (Frankl & Maehara, 1988), these vectors are likely to be unique, have constant norm, and be almost equidistant. We use the same $\mathbf{X}_h^0$ across all graphs and keep it fixed during training, effectively creating unique and consistent positional encodings for HR nodes. Combining it all together, we get the following message passing update:

$$\mathbf{X}_b = \begin{pmatrix} \tilde{\mathbf{X}}_l \\ \hat{\mathbf{X}}_h \end{pmatrix} = GNN_b(\mathbf{X}_b^0, \mathbf{A}_b) \quad s.t. \quad \mathbf{X}_b^0 = \begin{pmatrix} \hat{\mathbf{X}}_l \\ \mathbf{X}_h^0 \end{pmatrix} \tag{3}$$

where, $GNN_b : \mathbb{R}^{(n_l + n_h) \times d_l} \mapsto \mathbb{R}^{(n_l + n_h) \times d'}$. Unlike linear combination, message passing allows the node feature dimension to change.

**Node permutation invariance:** In section A.1, we prove that the linear algebraic and bipartite linear combination techniques are not invariant to LR node permutation, while bipartitie message passing is invariant.

### 3.1 REFINING HR NODE FEATURES

Our bipartite graph formulation initializes HR nodes directly from LR nodes but lacks interaction among the HR nodes themselves. To address this, we refine $\hat{\mathbf{X}}_h$ by incorporating intra-graph interactions using $\hat{\mathbf{X}}_h = GNN_{refine}(\hat{\mathbf{X}}_h, \mathbf{A}_h^{ref})$, where $GNN_{refine} : \mathbb{R}^{n_h \times d'} \mapsto \mathbb{R}^{n_h \times d''}$ and requires an adjacency matrix $\mathbf{A}_h^{ref} \in \mathbb{R}^{n_h \times n_h}$ as input. $\mathbf{A}_h^{ref}$ defines the HR computational domain, determining which nodes influence others during refinement. It differs from the predicted HR connectivity $\hat{\mathbf{A}}_h$, which is derived from $\hat{\mathbf{X}}_h$ as a downstream task. Below, we explore two ways to define this computational domain:

**Fixed Computation Domain:** A straightforward way is to assume a fully connected computational domain, allowing each HR node to interact with all others. Therefore, $\mathbf{A}_h^{ref}$ is defined as $\mathbf{A}_h^{ref} = \mathbf{1} - \mathbf{I}$, where $\mathbf{1} \in \mathbb{R}^{n_h \times n_h}$ is an all-ones matrix and $\mathbf{I} \in \mathbb{R}^{n_h \times n_h}$ is the identity matrix.

**Learnable Computation Domain:** Inspired by Zaripova et al. (2023), we propose learning $\mathbf{A}_h^{ref}$ by generating additional HR node features $\mathbf{X}_h^{ref}$. For bipartite linear combination: $\mathbf{X}_h^{ref} = \mathbf{W}_{ref}^b \hat{\mathbf{X}}_l$. For bipartite message passing: $\begin{pmatrix} \mathbf{X}_l^{ref} \\ \mathbf{X}_h^{ref} \end{pmatrix} = GNN_{b\_ref}(\mathbf{X}_b^0, \mathbf{A}_b)$ s.t. $GNN_{b\_ref} : \mathbb{R}^{n_l \times d} \mapsto \mathbb{R}^{n_h \times d_{ref}}$. Using these features, we compute $\mathbf{A}_h^{ref}$ as:

$$\hat{\mathbf{A}}_h^{ref} = \sigma(\mathbf{X}_h^{ref} \cdot \mathbf{X}_h^{ref\,T})$$
$$\mathbf{A}_h^{ref} = \mathbf{A}_h^{ref} = \hat{\mathbf{A}}_h^{ref} \odot H(\hat{\mathbf{A}}_h^{ref} - 0.5) \tag{4}$$

where, $\sigma$ is the sigmoid function and $H(x)$ is the Heaviside step (1 if $x \geq 0$, 0 otherwise).

### 3.2 GNN ARCHITECTURE

In this section, we present our GNN architecture for graph super resolution, combining the above techniques. Our super-resolution operator $\mathcal{S}$ predicts HR adjacency matrix $\hat{\mathbf{A}}_h$ from the LR features $\mathbf{X}_l$ and adjacency matrix $\mathbf{A}_l$ as: $\hat{\mathbf{A}}_h = \mathcal{S}(\mathbf{X}_l, \mathbf{A}_l; \theta)$, where $\theta$ represents learnable parameters. $\mathcal{S}$ can be decomposed into four main components:

***Part 1: Low resolution node representation learning***

We first embed $\mathbf{X}_l$ into a feature space more conducive to our task using a Graph Transformer Block (GTB), which consists of a Graph Transformer Layer followed by Graph Normalization (see section B.2): $\hat{\mathbf{X}}_l = \rho(\text{GTB}_1(\mathbf{X}_l, \mathbf{A}_l))$, where $\rho$ is the ReLU non-linearity.

***Part 2: Super-resolving the number of nodes***

We create HR node features from learned LR node features using one of three techniques:

1. Linear Algebraic method: $\hat{\mathbf{X}}_h = \hat{\mathbf{X}}_l^T$

2. Bipartite Linear Combination: $\hat{\mathbf{X}}_h = \mathbf{W}^b \hat{\mathbf{X}}_l$

3. Bipartite Message Passing: $\begin{pmatrix} \tilde{\mathbf{X}}_l \\ \hat{\mathbf{X}}_h \end{pmatrix} = \rho(\text{GTB}_2(\begin{pmatrix} \hat{\mathbf{X}}_l \\ \mathbf{X}_h^0 \end{pmatrix}, \mathbf{A}_b))$

where, $\mathbf{W}^b$ is a learnable weight matrix, $\mathbf{X}_h^0$ is the randomly initialized embedding for HR nodes, and $\mathbf{A}_b$ is the bipartite adjacency matrix.

***Part 3: (Optional) High resolution node representation learning***

To further refine the HR node features, we perform message passing on the HR graph using either the fixed or learnable computational domain from section 3.1: $\hat{\mathbf{X}}_h = \rho(\text{GTB}_3(\hat{\mathbf{X}}_h, \mathbf{A}_h^{ref}))$

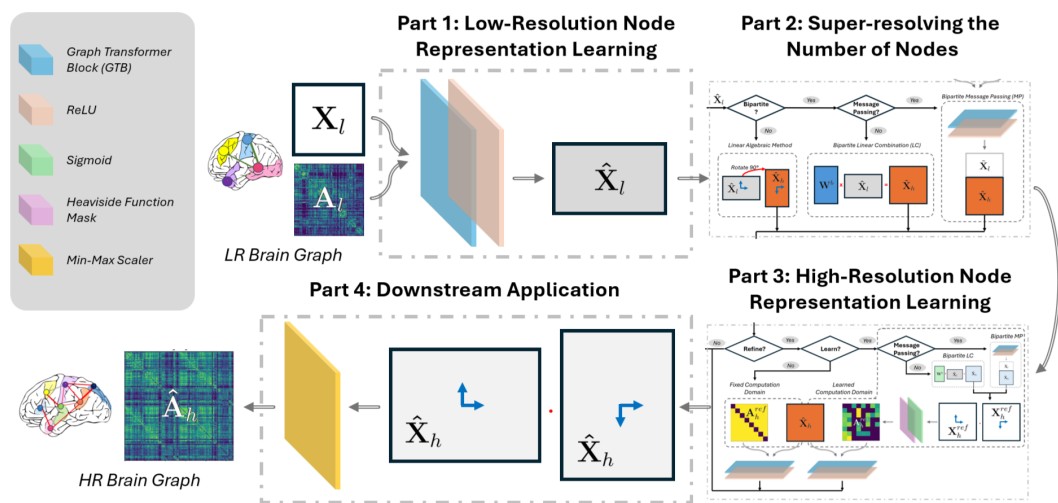

Figure 1: Overview for our super-resolution operator $\mathcal{S}$. Please refer to figure 7 for **Part 2** and **Part 3**.

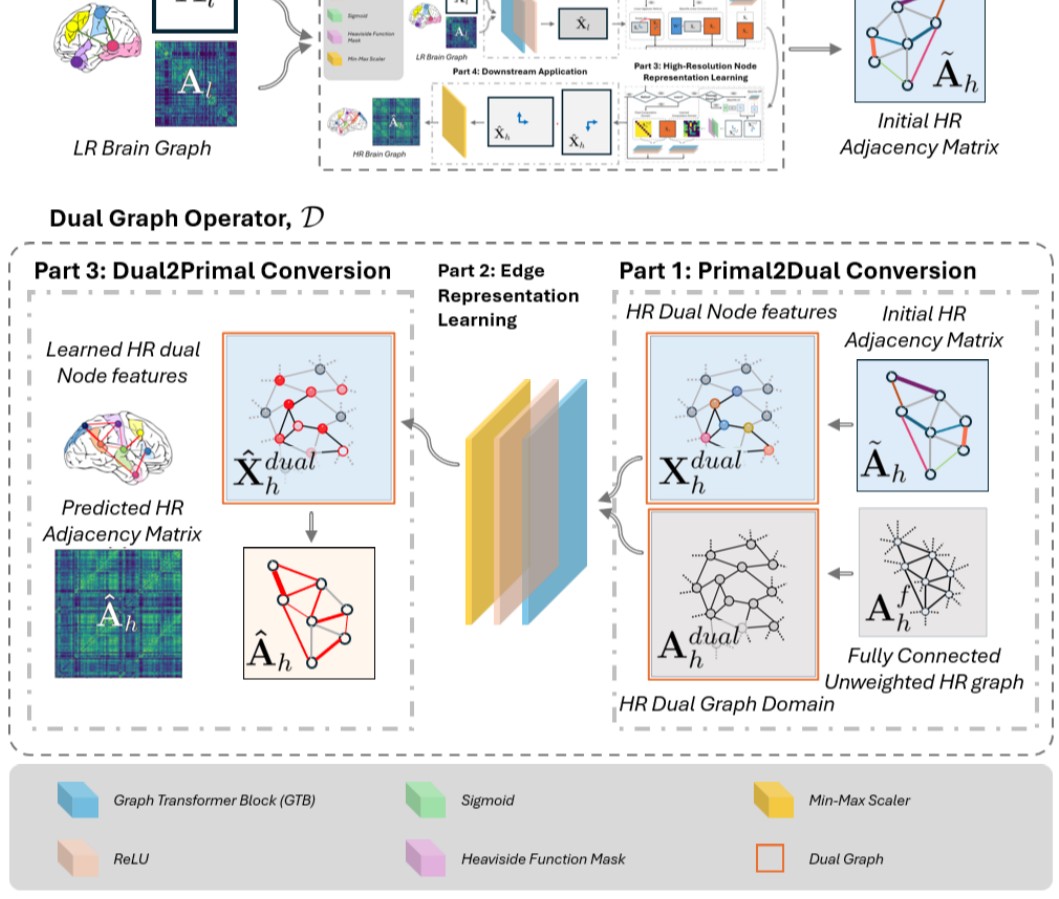

Figure 2: Overview of graph super-resolution framework using our Dual Graph Operator $\mathcal{D}$.

***Part 4: Downstream application***

Our primary application is predicting the HR connectivity matrix $\hat{\mathbf{A}}_h$ which is obtained by taking the dot product of learned HR node features $\hat{\mathbf{X}}_h$ and scaling values to $[0, 1]$ using $\beta$: $\hat{\mathbf{A}}_h = \beta(\hat{\mathbf{X}}_h \cdot \hat{\mathbf{X}}_h^T)$

Overall, this architecture flexibly combines various techniques to create a graph super-resolution framework suitable for different problems and computational requirements.

# 4   PROPOSED DEFEND FRAMEWORK

To enhance the edge modeling capacity of $\mathcal{S}$, we introduce a dual graph formulation which creates an invertible mapping between edges of our HR graph and nodes of a newly created dual graph:

---

**Dual Graph Formulation**

Given a simple undirected graph $\mathcal{G} = (\mathcal{V}, \mathcal{E}, \mathbf{A})$, known as the primal graph, its dual graph $\mathcal{G}' = (\mathcal{V}', \mathcal{E}', \mathbf{A}')$ is given as follows:

1. Each edge $(i, j) \in \mathcal{E}$ in $\mathcal{G}$ corresponds to a node $(i, j) \in \mathcal{V}'$ in $\mathcal{G}'$.

2. Two dual nodes $(i, j)$ and $(k, l)$ in $\mathcal{V}'$ are connected by an edge in $\mathcal{G}'$ if and only if the corresponding edges in $\mathcal{G}$ share a common node i.e. the condition $i = k$ or $i = l$ or $j = k$ or $j = l$ is satisfied.

3. Let $p$ and $q$ be the indices of the dual nodes $(i, j)$ and $(k, l)$ in $\mathcal{V}'$, respectively. Then, the adjacency matrix $\mathbf{A}'$ of the dual graph $\mathcal{G}'$ is defined as $\mathbf{A}'_{pq} = \mathbf{A}'_{qp} = 1$ if and only if $(i, j)$ and $(k, l)$ are connected as defined in 2. above; otherwise $\mathbf{A}'_{pq} = \mathbf{A}'_{qp} = 0$.

---

Above formulation retains all structural information of the primal graph. By treating edges as nodes, it permits direct application of node-based GNN layers for edge representation learning. This formulation can be extended to simple directed graphs by letting $(i, j) \in \mathcal{E}$ be an ordered set and connecting dual nodes $(i, j)$ and $(k, l)$ if they share a common node and a common direction. The resulting dual graphs are known as line (di)graphs or adjoint graphs in graph theory (Gross et al., 2018). For detailed computational analysis, please refer to section A.2.

## 4.1   THEORETICAL ANALYSIS

We present below for edge representation learning (see section A.3.2 and A.3.3 for their proof):

**Proposition 1:** Message passing in the edge space is more effective at modeling edge features compared to traditional message passing in the node space.

**Corollary 1:** Message passing in the edge space is more effective at learning graph topology compared to traditional message passing in the node space.

## 4.2   GNN ARCHITECTURE

We introduce a dual graph operator $\mathcal{D}$ that complements the super-resolution operator $\mathcal{S}$ by refining the connectivity matrix in the edge space via message passing on the HR dual graphs as $\hat{\mathbf{A}}_h = \mathcal{D}(\mathcal{S}(\mathbf{X}_l, \mathbf{A}_l, \theta), \omega)$, where $\omega$ are parameters of $\mathcal{D}$. $\mathcal{D}$ is decomposed into following parts:

***Part 1: Primal to Dual Conversion***

Let $\tilde{\mathbf{A}}_h = \mathcal{S}(\mathbf{X}_l, \mathbf{A}_l; \theta)$ be the initial HR adjacency matrix predicted by $\mathcal{S}$. Since the actual HR graph is unknown, we initialize the dual graph using a fully connected HR graph $\mathcal{G}_h^f = (\mathcal{V}_h^f, \mathcal{E}_h^f, \mathbf{A}_h^f)$ to account for all possible edges. We then construct the corresponding dual graph $\mathcal{G}_h^{dual} = (\mathcal{V}_h^{dual}, \mathcal{E}_h^{dual}, \mathbf{A}_h^{dual})$, where each edge $(i, j) \in \mathcal{E}_h^f$ maps to a node $k \in \mathcal{V}_h^{dual}$ via an invertible mapping $\phi$. We initialize the dual graph's feature matrix $\mathbf{X}_h^{dual}$ from $\tilde{\mathbf{A}}_h$ as $(\mathbf{X}_h^{dual})_k = (\tilde{\mathbf{A}}_h)ij$.

***Part 2: Edge Representation learning***

Using the GTB from section B.2, we perform message passing on the dual graph as: $\hat{\mathbf{X}}_h^{dual} = \beta(\rho(\text{GTB}_5(\mathbf{X}_h^{dual}, \mathbf{A}_h^{dual})))$, where $\rho$ is the ReLU non-linearity and $\beta$ is a scaling function.

***Part 3: Dual to Primal Conversion***

Finally, we convert the refined dual features $\hat{\mathbf{X}}_h^{dual}$ back to the HR adjacency matrix $\hat{\mathbf{A}}_h$ as follows:

$$(\hat{\mathbf{A}}_h)_{ij} = (\hat{\mathbf{A}}_h)_{ji} = \begin{cases} (\hat{\mathbf{X}}_h^{dual})_k & i \neq j \\ 0 & i = j \end{cases} \tag{5}$$

## 5 EXPERIMENTS

### 5.1 PHYSICS-INSPIRED DUMMY DATASET

To compare node v/s edge representation learning and empirically verify Proposition 1, we create datasets based on interacting particle systems, where each particle represents a node and have some mass and 2D position: *D1* (grid graph with random masses), *D2* (random graph with uniform mass), and *D3* (random graph with random masses). Edge values are derived using different functions: *E1* (inverse square law), *E2* (asymmetric rational), *E3* (symmetric quadratic), *E4* (symmetric polynomial), and *E5* (asymmetric quadratic). Combined, they cover a broad spectrum of scenarios where each representation learning framework succeeds/ struggles (see section C.1). We evaluate them using four models, two each for node and edge representation learning (see section D.1 and E.1 for model description and experimental set-up, respectively). Table 1 and 2 summarizes our results which are in line with expectations. For example, for *E1*, node-based models outperform edge-based ones on *D1* where edge value becomes dot product between masses, edge-based excel on *D2* as inverse square term dominates, and both perform comparably on *D3* where the dot product term compensates inverse square (see section F.1 for detailed performance analysis).

Table 1: Test MAE between true and predicted edge value for *E1* (inverse square law).

| Dataset | Node | Node Large | Edge | Edge Dual |
|---------|------|------------|------|-----------|
| *D1* | $\mathbf{0.869 \pm 0.032}$ | $\mathbf{1.136 \pm 0.899}$ | $2.371 \pm 2.087$ | $1.565 \pm 1.317$ |
| *D2* | $41.176 \pm 25.567$ | $39.525 \pm 28.190$ | $\mathbf{33.266 \pm 16.387}$ | $\mathbf{38.221 \pm 23.984}$ |
| *D3* | $13.499 \pm 9.805$ | $\mathbf{9.012 \pm 5.058}$ | $\underline{8.696 \pm 5.444}$ | $10.873 \pm 5.928$ |

Table 2: Test MAE between true and predicted edge value for *D3* (random graph w/ random masses).

| Edge Function | Node | Node Large | Edge | Edge Dual |
|---------------|------|------------|------|-----------|
| *E1* | $13.499 \pm 9.805$ | $\mathbf{9.012 \pm 5.058}$ | $\mathbf{8.696 \pm 5.444}$ | $10.873 \pm 5.928$ |
| *E2* | $26.611 \pm 9.176$ | $\mathbf{26.304 \pm 5.800}$ | $\mathbf{24.702 \pm 6.480}$ | $26.991 \pm 10.280$ |
| *E3* | $0.305 \pm 0.014$ | $0.325 \pm 0.037$ | $\underline{0.196 \pm 0.182}$ | $\mathbf{0.249 \pm 0.073}$ |
| *E4* | $\underline{0.485 \pm 0.039}$ | $0.663 \pm 0.391$ | $0.640 \pm 0.710$ | $\mathbf{0.639 \pm 0.275}$ |
| *E5* | $\underline{0.637 \pm 0.036}$ | $0.898 \pm 0.500$ | $0.821 \pm 0.934$ | $\mathbf{0.779 \pm 0.419}$ |

### 5.2 TRADITIONAL GRAPH GENERATION DATASET

To evaluate our GNN architectures across diverse graph topologies and LR-HR relationships, we generate twelve simulated datasets using three traditional models: Stochastic Block Model (SBM) for community structures, Barabási-Albert (BA) for scale-free networks, and Watts-Strogatz (WS) for small-world graphs. These models simulate HR graphs while the corresponding LR graphs are created using TopK pooling based on four node metrics $metric_{topK}$: Node Degree Centrality, Betweenness Centrality, Clustering Coefficient, and Participation Coefficient (see section C.2).

We benchmark fourteen ablated versions (section D.2) of our frameworks. Table 3 presents results for the WS datasets, grouping models into six categories with top-performing results for each: LA (Linear Algebraic Method), Bi-SR$_{LC}$ (Bipartite Linear Combination), Bi-SR$_{MP}$ (Bipartite Message

Passing), and their variations with the dual graph operator $\mathcal{D}$. See section F.2 for detailed analysis on all datasets. Despite deceptively simple scenarios (e.g., LR graphs missing clusters present in HR graphs), our bipartite graph formulation consistently outperforms the linear algebraic method. Specifically, Bi-SR$_{MP}$ outperforms Bi-SR$_{LC}$ on BA and WS datasets, with both performing comparably on SBM. The dual graph operator $\mathcal{D}$ offers no additional improvement, possibly due to edge representation learning's limited advantage for small graphs where node based models may suffice.

Table 3: Test MAE on $\mathbf{E}_h$ for the WS datasets. Each columns gives a $metric_{topK}$ dataset.

| Model | Degree | Betweenness | Clustering | Participation |
|---|---|---|---|---|
| LA | $2.179 \pm 0.132$ | $2.070 \pm 0.061$ | $2.128 \pm 0.053$ | $2.104 \pm 0.100$ |
| Bi-SR$_{LC}$ | $\mathbf{1.989 \pm 0.006}$ | $\mathbf{2.007 \pm 0.012}$ | $\mathbf{2.002 \pm 0.031}$ | $\mathbf{2.022 \pm 0.027}$ |
| Bi-SR$_{MP}$ | $\underline{1.998 \pm 0.020}$ | $\underline{2.002 \pm 0.005}$ | $\underline{1.994 \pm 0.030}$ | $\underline{2.010 \pm 0.025}$ |
| Dual LA | $2.149 \pm 0.011$ | $2.879 \pm 1.197$ | $2.156 \pm 0.027$ | $2.206 \pm 0.085$ |
| Dual Bi-SR$_{LC}$ | $2.027 \pm 0.040$ | $2.007 \pm 0.016$ | $2.009 \pm 0.024$ | $2.303 \pm 0.199$ |
| Dual Bi-SR$_{MP}$ | $2.022 \pm 0.048$ | $2.039 \pm 0.038$ | $2.083 \pm 0.122$ | $2.041 \pm 0.049$ |

## 5.3 BRAIN GRAPH DATASET

Using the publicly available SLIM dataset (Liu et al., 2017), we generate the LR-HR brain graph pairs using Dosenbach parcellated and Shen parcellated functional connectomes, respectively (section C.3). We compare sixteen models: fourteen ablated versions of our frameworks, an adapted version of the current state-of-the-art IMANGraphNet (Mhiri et al., 2021) (modified to address OOM error), and a new autoencoder baseline inspired by image super-resolution methods (section D.3).

Table 4: Performance on Brain Graph Dataset. Columns give test MAE across evaluation measures.

| Model | $\mathbf{A}_h$ $(10^1)$ | Betweenness $(10^4)$ | Closeness $(10^1)$ | Eigenvector $(10^3)$ |
|---|---|---|---|---|
| IMAN$_{adapted}$ | $1.725 \pm 0.074$ | $7.695 \pm 0.159$ | $1.590 \pm 0.028$ | $7.507 \pm 0.096$ |
| AutoEncoder | $1.381 \pm 0.062$ | $7.608 \pm 0.204$ | $1.520 \pm 0.025$ | $7.179 \pm 0.083$ |
| LA | $\mathbf{1.350 \pm 0.066}$ | $7.562 \pm 0.152$ | $1.513 \pm 0.033$ | $7.155 \pm 0.124$ |
| Bi-SR$_{LC}$ | $1.507 \pm 0.051$ | $7.693 \pm 0.159$ | $1.590 \pm 0.028$ | $7.506 \pm 0.096$ |
| Bi-SR$_{MP}$ | $1.428 \pm 0.052$ | $7.588 \pm 0.156$ | $1.551 \pm 0.039$ | $7.325 \pm 0.127$ |
| Dual LA | $1.458 \pm 0.153$ | $5.888 \pm 1.914$ | $1.133 \pm 0.442$ | $7.360 \pm 0.957$ |
| Dual Bi-SR$_{LC}$ | $1.515 \pm 0.293$ | $\mathbf{5.567 \pm 2.235}$ | $\mathbf{0.812 \pm 0.123}$ | $\mathbf{6.736 \pm 1.172}$ |
| Dual Bi-SR$_{MP}$ | $\underline{1.373 \pm 0.039}$ | $5.742 \pm 0.913$ | $1.046 \pm 0.128$ | $\underline{6.379 \pm 0.276}$ |

| Model | Degree $(10^0)$ | Participation $(10^1)$ | Clustering $(10^2)$ | Small Worldness $(10^2)$ |
|---|---|---|---|---|
| IMAN$_{adapted}$ | $54.778 \pm 1.170$ | $6.850 \pm 0.091$ | $14.006 \pm 0.318$ | $8.360 \pm 0.243$ |
| AutoEncoder | $51.697 \pm 1.038$ | $5.552 \pm 1.450$ | $14.193 \pm 0.437$ | $8.260 \pm 0.336$ |
| LA | $51.555 \pm 1.458$ | $5.255 \pm 0.883$ | $14.128 \pm 0.286$ | $8.126 \pm 0.289$ |
| Bi-SR$_{LC}$ | $54.771 \pm 1.170$ | $6.836 \pm 0.096$ | $14.003 \pm 0.318$ | $8.350 \pm 0.244$ |
| Bi-SR$_{MP}$ | $53.324 \pm 1.650$ | $5.090 \pm 0.837$ | $13.916 \pm 0.272$ | $8.254 \pm 0.243$ |
| Dual LA | $38.991 \pm 13.900$ | $3.401 \pm 3.172$ | $11.953 \pm 5.235$ | $5.873 \pm 3.221$ |
| Dual Bi-SR$_{LC}$ | $\mathbf{31.948 \pm 5.635}$ | $\mathbf{1.330 \pm 0.159}$ | $\mathbf{7.779 \pm 2.068}$ | $\mathbf{3.886 \pm 1.847}$ |
| Dual Bi-SR$_{MP}$ | $37.298 \pm 2.567$ | $1.440 \pm 0.233$ | $10.714 \pm 2.245$ | $\underline{5.322 \pm 1.068}$ |

We evaluate performance across eight measures: the MAE between the true and predicted $\mathbf{A}_h$, and the MAE of seven topological metrics capturing different aspects of brain connectivity. As shown in table 4, our dual graph formulation outperforms other methods across all topological metrics while

being competitive on $\mathbf{A}_h$ MAE. In the bipartite graph formulation, message passing outperforms linear combination, but this performance difference diminishes on supplementing with $\mathcal{D}$, possibly because $\mathcal{D}$ provides a powerful and robust edge learning approach that uplifts the performance of linear combination models. Our bipartite graph formulation does not improve over the linear algebraic method for this specific dataset. Finally, extensive sensitivity analysis (see section F.4) indicates that bipartite message passing is robust to variations in HR node initialization.

## 6 CONCLUSION

In this paper, we formalize graph super-resolution and tackle key limitations of existing work by introducing two novel frameworks, Bi-SR and DEFEND. Bi-SR is the first known framework to perform node super-resolution in a structurally consistent manner, while DEFEND provides a simple graph reformulation to perform edge representation learning using traditional node-based GNNs. Through extensive theoretical and empirical analysis, we demonstrate the superior performance and versatility of these frameworks, especially to ensure topological fidelity in generated graphs. We posit our work as general graph super-resolution frameworks. Therefore, as a future work, it would be worthwhile to explore how these can be adopted and optimized for domain-specific applications. Moreover, we observe that the relative performance of each framework depends on the scale of the graph. Therefore, it would be interesting to perform scaling analysis to understand this trade-off.

## 7 REPRODUCIBILITY STATEMENT

The code for running all experiments is attached under supplementary material to facilitate reproducibility. The appendix provides detailed proofs and derivations for all theoretical results, a comprehensive description of the data generation procedures for each dataset, the experimental setup, and an in-depth analysis of the results.

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

## A THEORETICAL ANALYSIS

### A.1 NODE PERMUTATION-INVARIANCE OF BI-SR

In this section, we compare the behavior of different super-resolution methods under the action of node permutation on the LR graph $\mathcal{G}_l$. Recall from Section 2.3 that the Graph Neural Networks (GNNs) considered in this work are node permutation equivariant, i.e.:

$$GNN(\mathbf{PX}, \mathbf{PAP}^T) = \mathbf{P}(GNN(\mathbf{X}, \mathbf{A})) \qquad (6)$$

where, $\mathbf{P}$ is the node permutation matrix. When applying $\mathbf{P} \in \mathbb{R}^{n_l \times n_l}$ to the LR nodes in $\mathcal{G}_l$, it changes the node feature matrix as follows:

$$\hat{\mathbf{X}}_l = GNN(\mathbf{X}_l, \mathbf{A}_l) \quad : \text{prior to application of } \mathbf{P} \qquad (7)$$

$$\hat{\mathbf{X}}_l^P = GNN(\mathbf{PX}_l, \mathbf{PA}_l\mathbf{P}^T) = \mathbf{P}\hat{\mathbf{X}}_l \quad : \text{after applying } \mathbf{P} \qquad (8)$$

For brevity, let us denote the super-resolution process as:

$$\hat{\mathbf{X}}_h = SRMethod(\hat{\mathbf{X}}_l) \qquad (9)$$

Our aim is to analyze whether the $SRMethod$ is invariant to $\mathbf{P}$ i.e., whether the following condition is satisfied:

$$\hat{\mathbf{X}}_h^P = SRMethod(\mathbf{P}\hat{\mathbf{X}}_l) = \hat{\mathbf{X}}_h \qquad (10)$$

Now, let's analyze different super-resolution techniques:

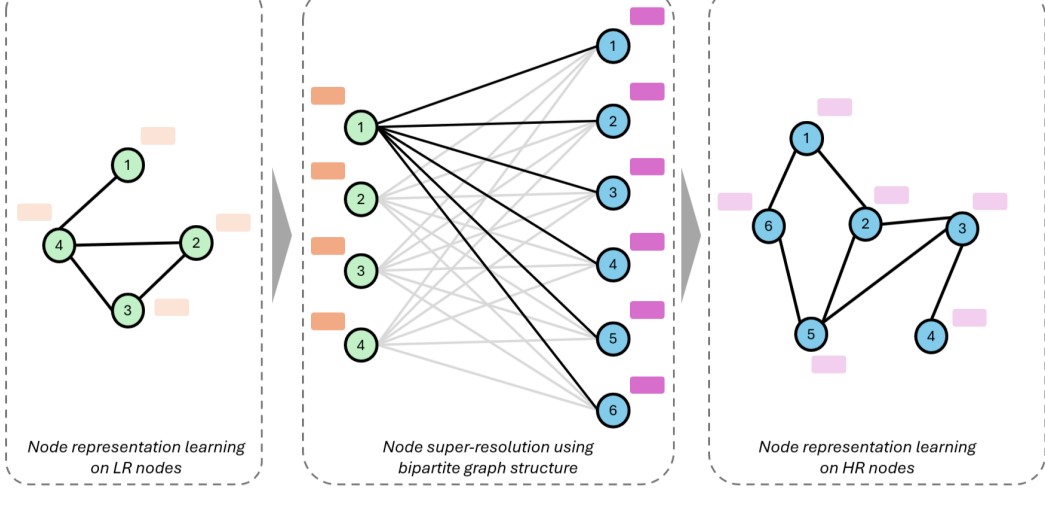

Figure 3: Bipartite Graph Formulation

1. **Linear Algebraic Method**

   Prior to application of $\mathbf{P}$:

   $$\hat{\mathbf{X}}_h = \hat{\mathbf{X}}_l^T \tag{11}$$

   After applying $\mathbf{P}$:

   $$\hat{\mathbf{X}}_h^P = (\mathbf{P}\hat{\mathbf{X}}_l)^T = \hat{\mathbf{X}}_l^T \mathbf{P}^T \neq \hat{\mathbf{X}}_h \tag{12}$$

2. **Bipartite Linear Combination**

   Prior to application of $\mathbf{P}$:

   $$\hat{\mathbf{X}}_h = \mathbf{W}^b \hat{\mathbf{X}}_l \tag{13}$$

   After applying $\mathbf{P}$:

   $$\hat{\mathbf{X}}_h^P = \mathbf{W}^b \mathbf{P}\hat{\mathbf{X}}_l \neq \hat{\mathbf{X}}_h \tag{14}$$

3. **Bipartite Message Passing**

   Prior to application of $\mathbf{P}$:

   $$\mathbf{X}_b = \begin{pmatrix} \tilde{\mathbf{X}}_l \\ \hat{\mathbf{X}}_h \end{pmatrix} = GNN_b(\mathbf{X}_b^0, \mathbf{A}_b) \quad s.t. \quad \mathbf{X}_b^0 = \begin{pmatrix} \hat{\mathbf{X}}_l \\ \mathbf{X}_h^0 \end{pmatrix} \tag{15}$$

   To analyze the effect of permutation, we first define the permutation matrix for the combined bipartite graph $\mathcal{G}_b$ as the block diagonal matrix:

   $$\mathbf{P}_b = \begin{pmatrix} \mathbf{P} & \mathbf{0} \\ \mathbf{0} & \mathbf{I} \end{pmatrix} \tag{16}$$

   where $\mathbf{0}$ are zero matrices of appropriate dimensions and $\mathbf{I} \in \mathbb{R}^{n_h \times n_h}$ is the identity matrix. This is equivalent to permuting the LR nodes in $\mathcal{G}_b$ according to $\mathbf{P}$ while keeping the HR nodes in $\mathcal{G}_b$ unchanged. Therefore, after applying $\mathbf{P}_b$:

   $$\mathbf{X}_b^P = GNN(\mathbf{P}_b\mathbf{X}_b^0, \mathbf{P}_b\mathbf{A}_b\mathbf{P}_b^T) = \mathbf{P}_b\mathbf{X}_b \tag{17}$$

   $$\mathbf{X}_b^P = \begin{pmatrix} \tilde{\mathbf{X}}_l^P \\ \hat{\mathbf{X}}_h^P \end{pmatrix} = \mathbf{P}_b \begin{pmatrix} \tilde{\mathbf{X}}_l \\ \hat{\mathbf{X}}_h \end{pmatrix} = \begin{pmatrix} \mathbf{P} & \mathbf{0} \\ \mathbf{0} & \mathbf{I} \end{pmatrix} \begin{pmatrix} \tilde{\mathbf{X}}_l \\ \hat{\mathbf{X}}_h \end{pmatrix} = \begin{pmatrix} \mathbf{P}\tilde{\mathbf{X}}_l \\ \hat{\mathbf{X}}_h \end{pmatrix} \tag{18}$$

From above, we can conclude that the linear algebraic and bipartite linear combination techniques are not invariant to LR node permutation, while the bipartite message passing method is invariant.

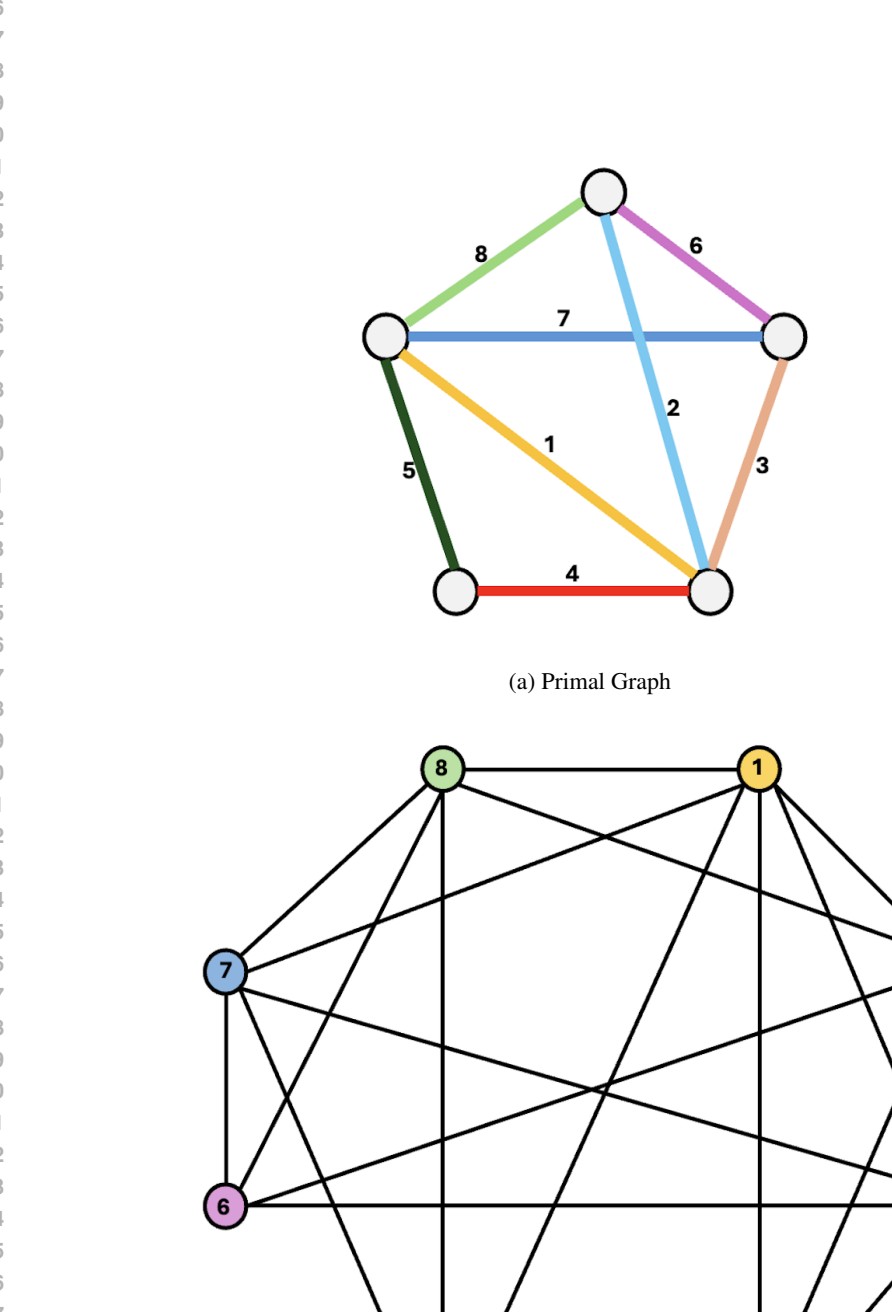

(a) Primal Graph

(b) Dual Graph

Figure 4: Example of a primal graph and its corresponding dual graph.

## A.2 COMPUTATIONAL PROPERTIES OF DUAL GRAPH FORMULATION

**Sparsity**: Let $n = |\mathcal{V}|$. In the worst case, our primal graph $\mathcal{G}$ is fully connected and the number of dual nodes grows quadratically: $|\mathcal{V}'| = |\mathcal{E}| = n(n-1)/2$. Since there are a maximum of $(n-1)$ edges originating from each node $i \in \mathcal{V}$, each edge $(i,j) \in \mathcal{E}$ has at most $m = 2(n-2)$ neighbors in the dual graph $\mathcal{G}'$. This results in a maximum of $|\mathcal{E}'| = |\mathcal{E}|m/2 = n(n-1)(n-2)/2$ dual edges, where the factor of 2 corrects double counting. Using this, we calculate the sparsity for dual adjacency matrix $\mathbf{A}'$ as:

$$sparsity \geq 1 - \frac{2 \times |\mathcal{E}'|}{|\mathcal{V}'|^2} = 1 - \frac{4(n-2)}{n(n-1)} \tag{19}$$

As seen in figure 5, this results in highly sparse dual graphs $\mathcal{G}'$ with worst case sparsity $> 90\%$ for $n \geq 39$. This allows us to leverage the in-built sparse matrix optimization in many deep learning libraries and significantly limit computational requirements.

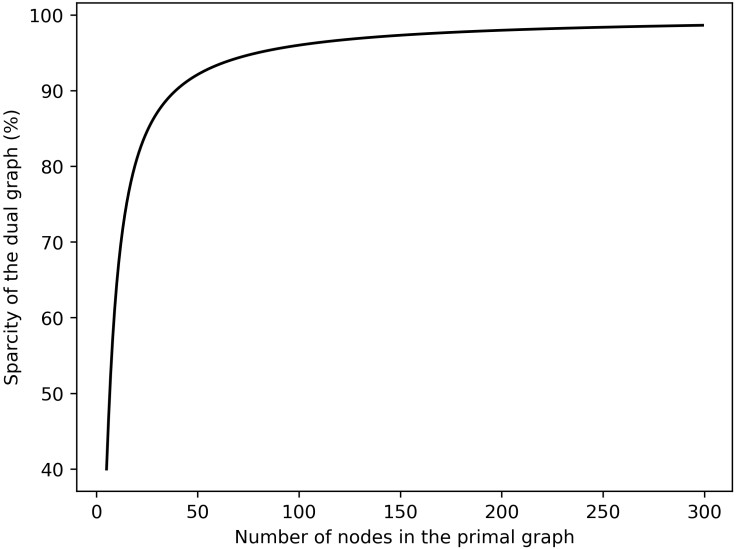

Figure 5: Worst case sparsity for dual graphs

**Receptive field**: As shown in figure 6, dual graphs have the same receptive field as primal graphs. However, they require half as many message passing operations as primal graphs to learn edge values. This results in a further reduction in computational requirement as message passing is an expensive operation.

**Dual node feature vectors**: Let $\mathbf{x}_i$ and $\mathbf{x}_j$ be node feature vectors for $i, j \in \mathcal{V}$. We initialize the node feature vector for dual node $(i,j) \in \mathcal{V}'$ as $\mathbf{e}_{ij}^0 = h(\mathbf{x}_i, \mathbf{x}_j)$, where $h$ is an arbitrary function acting on vectors. The most common form of $h$ is the concatenation operator $||$. As it is asymmetric, it is more suitable for directed graphs. For undirected graphs, a symmetric $h$ could be used such as vector summation, element wise product, dot product, etc.

## A.3 EDGE REPRESENTATION LEARNING

### A.3.1 UNIVERSAL FUNCTION APPROXIMATOR

A universal function approximator is a mathematical model capable of representing any continuous function to arbitrary accuracy, given sufficient resources. Hornik et al. (1989) demonstrated that multilayer feedforward networks possess this universal approximation property. We note the below lemma for universal function approximators:

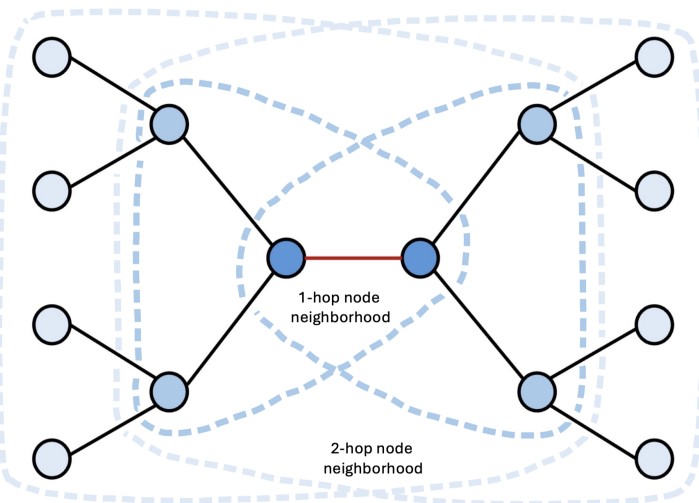

(a) Receptive field for node-based message passing

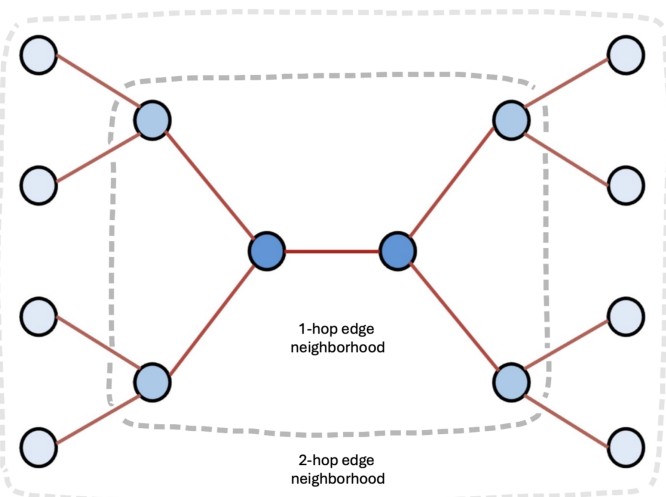

(b) Receptive field for edge-based message passing

Figure 6: Receptive fields for message passing in the node and edge space.

**Lemma 1**

A universal function approximator can be decomposed into a composition of multiple universal function approximators:

$$f_{uni}(\mathbf{x}) = f_{uni}^1(f_{uni}^2(...(f_{uni}^P(\mathbf{x})))) = f_{uni}^1 \circ f_{uni}^2 \circ ... \circ f_{uni}^P(\mathbf{x}) \tag{20}$$

From this lemma, we derive the following corollary:

**Corollary 2**

The concatenation of outputs from universal function approximators applied to separate inputs forms a subset of the output produced by applying a universal function approximator to the concatenated inputs:

$$f_{uni}([\mathbf{x}||\mathbf{y}]) = f_{uni}^1(f_{uni}^2([\mathbf{x}||\mathbf{y}]))) \supseteq [f_{uni}^3(\mathbf{x})||f_{uni}^4(\mathbf{y})] \tag{21}$$

where $||$ represents the concatenation operator.

**Proof of Corollary 2**: Let $\mathbf{x} \in \mathbb{R}^{d_x}$ and $\mathbf{y} \in \mathbb{R}^{d_y}$ be two input vectors. Consider universal function approximators $f_{uni}^1$ and $f_{uni}^2$ acting on $\mathbb{R}^{d_x + d_y}$, while $f_{uni}^3$ acts on $\mathbb{R}^{d_x}$ and $f_{uni}^4$ acts on $\mathbb{R}^{d_y}$. Restricting $f_{uni}^2$ to functions that apply $f_{uni}^3$ to the first $d_x$ dimensions gives:

$$f_{uni}^2([\mathbf{x}||\mathbf{y}]) \supseteq [f_{uni}^3(\mathbf{x})||\mathbf{y}] \tag{22}$$

Restricting $f_{uni}^1$ to functions that apply $f_{uni}^4$ to the last $d_y$ dimensions gives:

$$f_{uni}^1([\mathbf{w}||\mathbf{y}]) \supseteq [\mathbf{w}||f_{uni}^4(\mathbf{y})] \tag{23}$$

Finally, combining these results:

$$f_{uni}^1(f_{uni}^2([\mathbf{x}||\mathbf{y}]))) \supseteq [f_{uni}^3(\mathbf{x})||f_{uni}^4(\mathbf{y})] \tag{24}$$

### A.3.2 PROPOSITION

We propose the following for our edge representation learning framework:

**Proposition 1**

Message passing in the edge space is more effective at modeling edge features compared to traditional message passing in the node space.

**Proof of Proposition 1**:

***Traditional Message Passing in the node space***: Recall from section 2.3 that traditional message-passing in the node space consists of two key components:

1. Neighborhood aggregation:

$$\mathbf{z}_i^l = \beta^l \mathbf{x}_i^{l-1} + (1 - \beta^l) \sum_{j \in \mathcal{N}_i} \alpha_{ij}^l \mathbf{x}_j^{l-1} = g_n^a(\{\mathbf{x}_i^{l-1}\}_{\mathcal{N}}) \tag{25}$$

2. Node feature update:

$$\mathbf{x}_i^l = f_n(\mathbf{z}_i^l) = f_n \circ g_n^a(\{\mathbf{x}_i^{l-1}\}_{\mathcal{N}}) \tag{26}$$

By letting $f_n$ be a universal function approximator, we achieve a maximally powerful MPNN that generalizes to a wide range of GNN architectures, including GIN (Xu et al., 2018), GAT (Velickovic et al., 2017), and Graph Transformers (Shi et al., 2020). For brevity, let $\mathbf{x}_i = \mathbf{x}_i^{l-1}$, $\hat{\mathbf{x}}_i = \mathbf{x}_i^l$, $\{\mathbf{x}_i\}_{\mathcal{N}}$ be the set containing the feature vectors for node $i$ and all its neighbors, and $f_n^{mp} = f_n \circ g_n^a$. Then, the message passing update can be re-written as:

$$\hat{\mathbf{x}}_i = f_n^{mp}(\{\mathbf{x}_i\}_{\mathcal{N}}) \tag{27}$$

To learn edge features, we usually encode node features in a dot product space where the dot product corresponds to the edge feature value. Therefore:

$$\hat{e}_{ij} = \hat{\mathbf{x}}_i \cdot \hat{\mathbf{x}}_j = f_n^{mp}(\{\mathbf{x}_i\}_{\mathcal{N}}) \cdot f_n^{mp}(\{\mathbf{x}_j\}_{\mathcal{N}}) \tag{28}$$

***Message Passing in the edge space***: For edge representation learning, let's initialize edge feature vectors as $\mathbf{e}_{ij}^0 = [\mathbf{x}_i || \mathbf{x}_j]$, as is common practice. The message passing formulation for edges is then given as:

1. Neighborhood aggregation:

$$
\begin{aligned}
\mathbf{z}_{ij} = \beta \mathbf{e}_{ij}^0 + (1-\beta)[ &\sum_{k \in \mathcal{N}_i} \alpha_{(ij)(ik)} \mathbf{e}_{ik}^0 + \sum_{l \in \mathcal{N}_j} \alpha_{(ij)(lj)} \mathbf{e}_{lj}^0 \\
&+ \sum_{s \in \mathcal{N}_i} \alpha_{(ij)(si)} \mathbf{e}_{si}^0 + \sum_{r \in \mathcal{N}_j} \alpha_{(ij)(jr)} \mathbf{e}_{jr}^0)]
\end{aligned}
\tag{29}
$$

2. Edge feature update:

$$\hat{\mathbf{e}}_{ij} = f_e(\mathbf{z}_{ij}) \tag{30}$$

where $f_e$ is a universal function approximator.

***Relationship between node space and edge space message passing***: Expanding $\mathbf{e}_{ij}^0$ and simplifying equation 29:

$$
\begin{aligned}
\hat{\mathbf{e}}_{ij} = f_e(\beta [\mathbf{x}_i || \mathbf{x}_j] + (1-\beta)[ &\sum_{k \in \mathcal{N}_i} \alpha_{(ij)(ik)} [\mathbf{x}_i || \mathbf{x}_k] + \sum_{l \in \mathcal{N}_j} \alpha_{(ij)(lj)} [\mathbf{x}_l || \mathbf{x}_j] \\
&+ \sum_{s \in \mathcal{N}_i} \alpha_{(ij)(si)} [\mathbf{x}_s || \mathbf{x}_i] + \sum_{r \in \mathcal{N}_j} \alpha_{(ij)(jr)} [\mathbf{x}_j || \mathbf{x}_r])])
\end{aligned}
\tag{31}
$$

$$
\begin{aligned}
\hat{\mathbf{e}}_{ij} = f_e([&(\mathbf{c}_i^1 \mathbf{x}_i + \mathbf{c}_j^1 \mathbf{x}_j + \sum_{k \in \mathcal{N}_i} \mathbf{c}_k^1 \mathbf{x}_k + \sum_{l \in \mathcal{N}_j} \mathbf{c}_l^1 \mathbf{x}_l) \\
&|| (\mathbf{c}_i^2 \mathbf{x}_i + \mathbf{c}_j^2 \mathbf{x}_j + \sum_{k \in \mathcal{N}_i} \mathbf{c}_k^2 \mathbf{x}_k + \sum_{l \in \mathcal{N}_j} \mathbf{c}_l^2 \mathbf{x}_l)])
\end{aligned}
\tag{32}
$$

$$\hat{\mathbf{e}}_{ij} = f_e([g_1^a(\mathbf{x_i}, \mathbf{x_j}, \{\mathbf{x}_i\}_{\mathcal{N}}, \{\mathbf{x}_j\}_{\mathcal{N}}) || g_2^a(\mathbf{x_i}, \mathbf{x_j}, \{\mathbf{x}_i\}_{\mathcal{N}}, \{\mathbf{x}_j\}_{\mathcal{N}})]) \tag{33}$$

Let $g_1^{sub} \subseteq g_1^a$ and $g_2^{sub} \subseteq g_2^a$ be subsets of aggregation functions that zero out some inputs s.t. $g_1^{sub}(w,x,y,z) = g_1^{sub}(y)$ and $g_2^{sub}(w,x,y,z) = g_2^{sub}(z)$. Then:

$$
\begin{aligned}
\hat{\mathbf{e}}_{ij} &= f_e([g_1^a(\mathbf{x_i}, \mathbf{x_j}, \{\mathbf{x}_i\}_{\mathcal{N}}, \{\mathbf{x}_j\}_{\mathcal{N}}) || g_2^a(\mathbf{x_i}, \mathbf{x_j}, \{\mathbf{x}_i\}_{\mathcal{N}}, \{\mathbf{x}_j\}_{\mathcal{N}})]) \\
\hat{\mathbf{e}}_{ij} &\supseteq f_e([g_1^{sub}(\{\mathbf{x}_i\}_{\mathcal{N}}) || g_2^{sub}(\{\mathbf{x}_j\}_{\mathcal{N}})])
\end{aligned}
\tag{34}
$$

Applying Lemma 1:

$$f_e([g_1^{sub}(\{\mathbf{x}_i\}_{\mathcal{N}}) || g_2^{sub}(\{\mathbf{x}_j\}_{\mathcal{N}})]) = f_e^1 \circ f_e^2 \circ f_e^3([g_1^{sub}(\{\mathbf{x}_i\}_{\mathcal{N}}) || g_2^{sub}(\{\mathbf{x}_j\}_{\mathcal{N}})]) \tag{35}$$

Applying Corollary 1:

$$f_e^1 \circ f_e^2 \circ f_e^3([g_1^{sub}(\{\mathbf{x}_i\}_{\mathcal{N}}) || g_2^{sub}(\{\mathbf{x}_j\}_{\mathcal{N}})]) \supseteq f_e^1([f_e^4 \circ g_1^{sub}(\{\mathbf{x}_i\}_{\mathcal{N}}) || f_e^5 \circ g_2^{sub}(\{\mathbf{x}_j\}_{\mathcal{N}})]) \tag{36}$$

Let $f_e^{mp1} = f_e^4 \circ g_1^{sub}$ and $f_e^{mp2} = f_e^5 \circ g_2^{sub}$, then:

$$f_e^1([f_e^4 \circ g_1^{sub}(\{\mathbf{x}_i\}_{\mathcal{N}}) || f_e^5 \circ g_2^{sub}(\{\mathbf{x}_j\}_{\mathcal{N}})]) = f_e^1([f_e^{mp1}(\{\mathbf{x}_i\}_{\mathcal{N}}) || f_e^{mp2}(\{\mathbf{x}_j\}_{\mathcal{N}})]) \tag{37}$$

As $f_e^1$ is a universal function approximator, it can represent the function $f([x||y]) = x \cdot y$. Therefore:

$$f_e^1([f_e^{mp1}(\{\mathbf{x}_i\}_{\mathcal{N}}) || f_e^{mp2}(\{\mathbf{x}_j\}_{\mathcal{N}})]) \supseteq f_e^{mp1}(\{\mathbf{x}_i\}_{\mathcal{N}}) \cdot f_e^{mp2}(\{\mathbf{x}_j\}_{\mathcal{N}}) \tag{38}$$

If we restrict $f_e^{mp1}$ and $f_e^{mp2}$ s.t. $f_e^{mp1} = f_e^{mp2} = f_e^{mp}$, we get:

$$f_e^{mp1}(\{\mathbf{x}_i\}_{\mathcal{N}}) \cdot f_e^{mp2}(\{\mathbf{x}_j\}_{\mathcal{N}}) \supseteq f_e^{mp}(\{\mathbf{x}_i\}_{\mathcal{N}}) \cdot f_e^{mp}(\{\mathbf{x}_j\}_{\mathcal{N}}) \tag{39}$$

The right hand side is similar to the node space message passing equation 28. Combining it all together, we get:

$$f_e(\mathbf{z}_{ij}) \supseteq f_e^1([f_e^{mp1}(\{\mathbf{x}_i\}_{\mathcal{N}})||f_e^{mp2}(\{\mathbf{x}_j\}_{\mathcal{N}})]) \supseteq f_n^{mp}(\{\mathbf{x}_i\}_{\mathcal{N}}) \cdot f_n^{mp}(\{\mathbf{x}_j\}_{\mathcal{N}}) \tag{40}$$

Thus, we have shown that message passing in the edge space is at least as powerful as, and potentially more powerful than, traditional message passing in the node space for the task of learning edge features.

### A.3.3 COROLLARY

We derive below corollary for our edge representation learning framework:

> **Corollary 1**
>
> Message passing in the edge space is more effective at learning graph topology compared to traditional message passing in the node space.

**Proof of Corollary 1**: Graph topology can be succinctly represented by the adjacency matrix $\hat{\mathbf{A}}$. Let $\hat{\mathbf{A}}_{ij} = \hat{e}_{ij}$. Then, our corollary follows directly from Proposition 1.

## B GNN ARCHITECTURE

### B.1 GRAPH TRANSFORMER LAYER

Graph transformer layer extends the self-attention mechanism to graph data structure. Assuming $H$ attention heads, the representation for node $i$ is updated as:

$$\begin{aligned}
\mathbf{x}_i^l &= \mathbf{W}_0[\mathbf{z}_i^{l^1}||\mathbf{z}_i^{l^2}||...||\mathbf{z}_i^{l^H}] \\
\mathbf{z}_i^{l^h} &= \mathbf{W}_1^h\mathbf{x}_i^{l-1} + \sum_{j \in \mathcal{N}_i} \alpha_{ij}^h(\mathbf{W}_2^h\mathbf{x}_j^{l-1} + \mathbf{W}_6^h\mathbf{A}_{ij}) \\
\alpha_{ij}^h &= softmax\left(\frac{(\mathbf{W}_3^h\mathbf{x}_i^{l-1})^T(\mathbf{W}_4^h\mathbf{x}_j^{l-1} + \mathbf{W}_5^h\mathbf{A}_{ij})}{\sqrt{d}}\right)
\end{aligned} \tag{41}$$

where, $||$ is the concatenation operator, $d$ is the dimension of $\mathbf{x}_i^{l-1}$, and $\{\mathbf{W}_k^h|k \in \{1, 2, ..., 6\}, h \in \{1, 2, ..., H\}\} \cup \mathbf{W}_0$ are learnable parameters. We chose graph transformer as our primary message passing layer since the learned node features are more expressive than GCNConv (Kipf & Welling, 2016) while being more computationally efficient than NNConv (Simonovsky & Komodakis, 2017), two widely used layers in graph super-resolution.

### B.2 GRAPH TRANSFORMER BLOCK

Our architecture utilizes the Graph Transformer Layer (GTL) defined in section B.1 as the primary message passing mechanism. Each GTL is followed by Graph Normalization (GraphNorm) to stabilize and accelerate training. We define this combined operation as the Graph Transformer Block (GTB):

$$GTB = GraphNorm(GTL(\cdot)) \tag{42}$$

The GTB serves as the foundation for all GNNs in this work.

## C DATA GENERATION

### C.1 PHYSICS-INSPIRED DUMMY DATASET

In this work, we presented our edge representation learning framework and proposed theoretical justification for why edge space computations are more effective than node space computations to

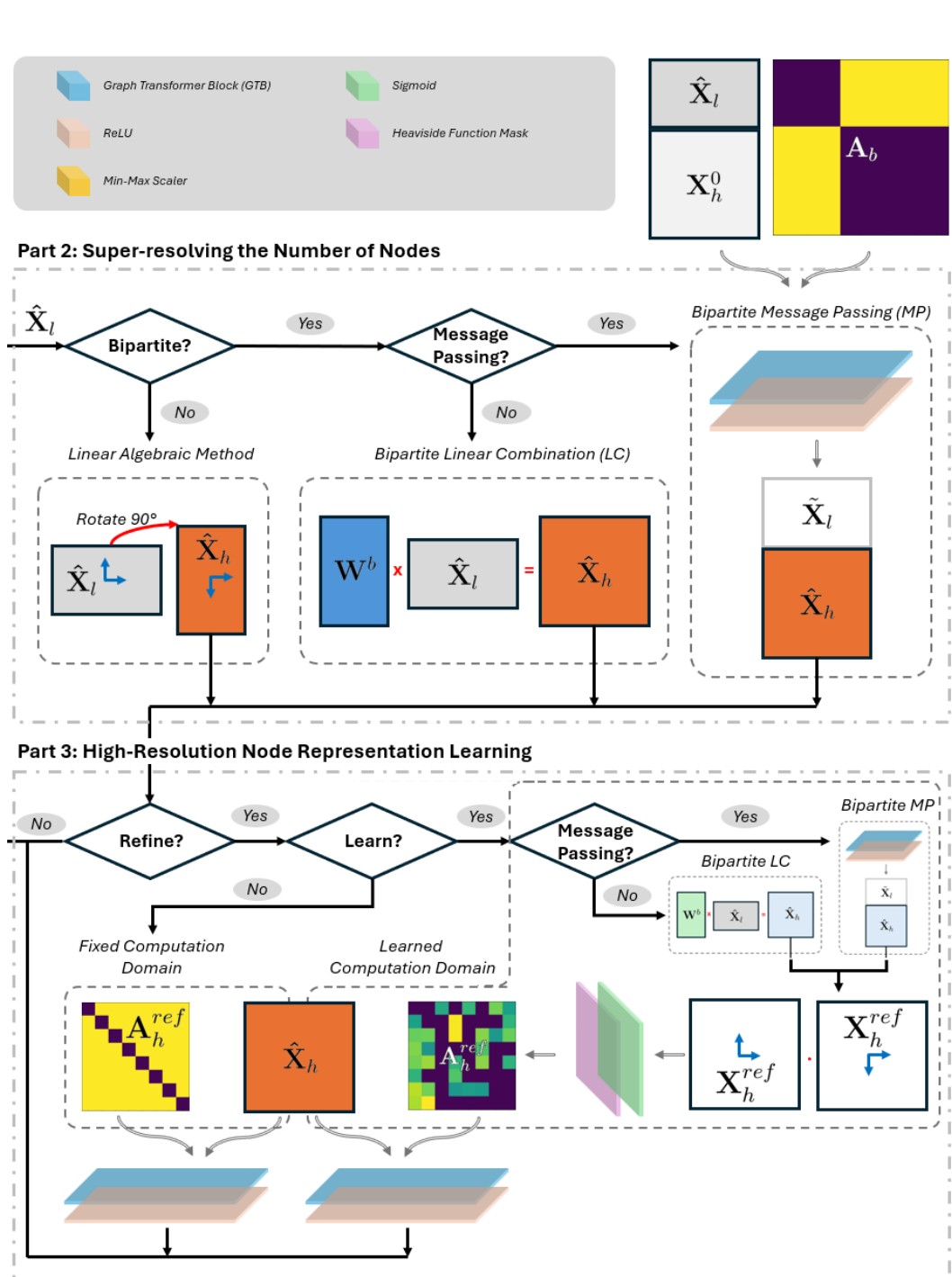

Figure 7: Overview of *Part 2* (Super-resolving the number of nodes) and *Part 3* (HR node representation learning) for the super-resolution operator S.

predict edge features. To empirically evaluate this claim, we simulate a dataset inspired by interacting particle systems in Physics. Specifically, we initialize a set of particles in 2D space and assign them masses. We then assume each particle to be a node in a graph and initialize the node feature vector as a combination of mass and position i.e. for the $i^{th}$ particle, its node feature vector $\mathbf{x}_i$ is $[m_i, x_i, y_i]$. We then create edges between these nodes based on some heuristic. Finally, we assign each edge a value which follows the inverse square law:

***E1: Inverse square law***

$$e_{ij} = \frac{Gm_i m_j}{r_{ij}^2}$$
$$r_{ij}^2 = (x_i - x_j)^2 + (y_i - y_j)^2 \tag{43}$$

We hypothesize that node representation learning models may be able to easily model the dot product term $m_i m_j$ while struggling to model the inverse distance term $1/r_{ij}^2$ which varies a lot and necessitates encoding the relative distance to each neighbor individually.

There are two variables for each particle: the mass $m_i$ and the position $(x_i, y_i)$. This gives us three non-trivial datasets:

- ***D1: Grid graph with random masses***

  Particles are laid out uniformly on a square grid with the masses drawn randomly from the uniform distribution $m_i \sim \mathcal{U}(0, 1)$. As its a grid graph, we keep $r_{ij} = 1.0$ for all edges. Therefore, $e_{ij} \propto m_i m_j$, and we expect the node representation learning model to perform well as they essentially perform dot product between learned node features to predict $e_{ij}$.

- ***D2: Random geometric graph with uniform mass***

  Particle positions are sampled uniformly from a unit square, and two nodes are connected if the distance between them is less than a given threshold $t$. Moreover, we keep $m_i = 1.0$ for all nodes to isolate the impact of distances. We expect the edge representation learning model to perform better as the edge values only depend on the relative distance between the nodes.

- ***D3: Random geometric graph with random mass***

  This provides a general case of the random geometric graphs in ***D2*** with $m_i \sim \mathcal{U}(0, 1)$. This represents a more realistic graph setting with multiple competing components s.t. the edge value depends both on the dot product of the masses and the relative distance.

While inverse square law provides an edge function that is difficult to model, we supplement our analysis with additional edge functions that highlight different aspects of node v/s edge representation learning:

- ***E2: Asymmetric rational function***

  $$e_{ij} = \frac{Am_i + Bm_j}{x_i^2 + y_j^2} \tag{44}$$

  The denominator depends on an asymmetric combination of absolute node positions, making it difficult to encode within a single node feature. In contrast, the numerator, while also asymmetric, should be easier to encode as it denotes a simple linear combination.

- ***E3: Symmetric quadratic function***

  $$e_{ij} = (x_i - x_j)^2 + (y_i - y_j)^2 + (m_i - m_j)^2 \tag{45}$$

  This represents the squared Euclidean distance between the two node features. Unlike the rational functions ***E1*** and ***E2***, we expect node based models to struggle since there are no compensating numerator terms to bring down the total loss.

- ***E4: Symmetric polynomial function***

  $$e_{ij} = x_i y_i m_i + x_j y_j m_j + x_i y_j + x_j y_i \tag{46}$$

  It may be possible to approximate this equation as a dot product between individual node features by projecting the node features to a higher dimensional sparse feature space. Therefore, we do not expect the node based models to struggle.

- **E5: Asymmetric quadratic function**

$$e_{ij} = x_i^2 + y_i^2 + m_j^2 \tag{47}$$

Even though this is asymmetric, it may possible to learn this function similar to **E4** by projecting the node features to non-linear higher dimensional space. Therefore, we do not expect the node based models to struggle.

## C.2 TRADITIONAL GRAPH SIMULATION DATASET

### C.2.1 HR GRAPH GENERATION

We employ three widely recognized graph generation models to create our HR graphs: (1) Stochastic Block Model (SBM) (Lee & Wilkinson, 2019) that generates community or clustered graphs; (2)Barabási-Albert (BA) Model (Barabási & Albert, 1999) that produces scale-free graphs; (3) Watts-Strogatz (WS) Model (Watts & Strogatz, 1998) that creates small-world graphs. Below, we discuss each of these models in detail:

- **Stochastic Block Model (SBM)**: The SBM generates graphs by partitioning nodes into multiple clusters and probabilistically connecting them. The generation process requires two key inputs:
  1. The number of nodes in each cluster: $\mathbf{c} = [n_1, n_2, ..., n_c]$, where $c$ is the number of clusters and $\sum_{i=1}^{c} n_i = |\mathcal{V}| = n$.
  2. The connection probability matrix: $\mathbf{P}^{sbm} \in \mathbb{R}^{c \times c}$, where $\mathbf{P}^{sbm}_{ij}$ defines the probability of connecting nodes in cluster $i$ with nodes in cluster $j$. The intra cluster probabilities $\mathbf{P}^{sbm}_{ij} |_{i=j}$ are usually higher than inter cluster probabilities $\mathbf{P}^{sbm}_{ij} |_{i \neq j}$ to create clusters.

  While the generated graphs are already stochastic, we further randomize above inputs to ensure a topologically diverse dataset. For this, we sample $c$ from $[c_{min}, c_{max}]$, initialize $\mathbf{P}^{sbm}_{ij} |_{i=j}$ from $[p_{min}^{intra}, p_{max}^{intra}]$, and initialize $\mathbf{P}^{sbm}_{ij} |_{i \neq j}$ from $[p_{min}^{inter}, p_{max}^{inter}]$. Moreover, we use the multinomial distribution to partition the $n$ nodes into $c$ clusters to ensure that no cluster ends up with very few nodes. Algorithm 2 provides the pseudocode for our simulation process and Figure 8a shows the variation of generated graphs with $\mathbf{P}^{sbm}$.

- **Barabási-Albert (BA) Model**: The BA model generates scale-free graphs by introducing preferential attachment during network growth. This means that each incoming node connects to an existing node with a probability based on its current degree. Therefore, nodes created early on are more likely to be connected to new nodes and continue growing into hubs. The simulated graphs show power-law degree distribution and mimic many real-world datasets such as social interactions, internet connectivity, etc. The generation process requires two user inputs:
  1. The total number of nodes in the graph $n$
  2. The number of edges $m$ to attach from each new node to existing nodes while growing the network

  Similar to the SBM model, this process is also stochastic and leads to different graphs. However, to ensure a higher topological diversity, we randomly sample $m$ from the range $[m_{min}, m_{max}]$. Algorithm 3 provides the pseudocode for our simulation process and Figure 8b shows the variation in graph structure with $m$.

- **Watts-Strogatz (WS) Model**: The WS model generates small-world graphs that possess high clustering and short average path lengths. This is done by initializing the graphs as a regular ring lattice where every node is connected to its $k$ nearest neighbors. Thereafter, it rewires each edge with a probability $p$ to introduce randomness. Small-world graphs provide a good mathematical model for numerous natural graphs such as neural networks and power grids where high clustering reflects high activity regions while short path lengths correspond to rapid signal transmission. The generation process requires three inputs:
  1. The total number of nodes in the graph $n$
  2. The number of nearest neighbor connections $k$ for the regular ring lattice
  3. The rewiring probability $p$

  As before, we supplement topological diversity of our graphs by randomizing above inputs. This involves sampling $k$ from $[k_{min}, k_{max}]$ to control initial lattice structure and sampling $p$ from

$[p_{min}, p_{max}]$ to control rewiring strength. Algorithm 4 gives the pseudocode for our simulation process and Figure 8c shows the variation with $p$.

---

**Algorithm 1** Generate LR graph and compute node and edge features

---

1: **Input:** $\mathcal{V}_h, \mathcal{E}_h, \mathbf{A}_h, metric_{topK}$
2: **Output:** $(\mathcal{G}_l, \mathcal{G}_h)$             $\triangleright$ (LR, HR) attributed graph pair
3: $\mathbf{X}_h = Node2Vec(\mathbf{A}_h)$        $\triangleright$ Generate node feature matrix for HR graph
4: $\mathbf{E}_h = PearsonCorr(\mathbf{X}_h)$       $\triangleright$ Generate edge feature tensor for HR graph
5: $\mathcal{G}_h = (\mathcal{V}_h, \mathcal{E}_h, \mathbf{A}_h, \mathbf{X}_h, \mathbf{E}_h)$
6: $\mathcal{V}_l, \mathcal{E}_l, \mathbf{A}_l = TopK(\mathcal{G}_h, metric_{topK})$     $\triangleright$ Create LR graph using TopK pooling
7: $\mathbf{X}_l = Node2Vec(\mathbf{A}_l)$        $\triangleright$ Generate node feature matrix for LR graph
8: $\mathbf{E}_l = PearsonCorr(\mathbf{X}_l)$       $\triangleright$ Generate edge feature tensor for LR graph
9: $\mathcal{G}_l = (\mathcal{V}_l, \mathcal{E}_l, \mathbf{A}_l, \mathbf{X}_l, \mathbf{E}_l)$
10: **return** $(\mathcal{G}_l, \mathcal{G}_h)$

---

**Algorithm 2** Data simulation using Stochastic Block Model (SBM)

---

1: **Input:** $N, n, c_{min}, c_{max}, p_{min}^{inter}, p_{max}^{inter}, p_{min}^{intra}, p_{max}^{intra}, metric_{topK}$
2: **Output:** $data$              $\triangleright$ List of (LR, HR) graph pairs
3: $data \leftarrow []$           $\triangleright$ Initialize an empty list to store graph pairs
4: **for** $l \leftarrow 1$ **to** $N$ **do**
5:     $c \sim \mathcal{U}(c_{min}, c_{max})$        $\triangleright$ Initialize the number of clusters in this graph
6:     $\mathbf{c} \sim Multinomial(n - c, 1/c) + 1$       $\triangleright$ Distribute $n$ nodes into $c$ clusters
7:     $\mathbf{P}^{sbm} \leftarrow \mathbf{0} \in \mathbb{R}^{c \times c}$       $\triangleright$ Initialize the connection probability matrix
8:     **for** $i \leftarrow 1$ **to** $c$ **do**
9:        **for** $j \leftarrow 1$ **to** $c$ **do**
10:           **if** $i = j$ **then**
11:             $\mathbf{P}_{ij}^{sbm} \sim \mathcal{U}(p_{min}^{intra}, p_{max}^{intra})$       $\triangleright$ Assign intra cluster probability
12:           **else**
13:             $\mathbf{P}_{ij}^{sbm} \sim \mathcal{U}(p_{min}^{inter}, p_{max}^{inter})$       $\triangleright$ Assign inter cluster probability
14:           **end if**
15:        **end for**
16:     **end for**
17:     $\mathcal{V}_h, \mathcal{E}_h, \mathbf{A}_h = SBM(\mathbf{c}, \mathbf{P})$        $\triangleright$ Create HR graph structure
18:     $(\mathcal{G}_l, \mathcal{G}_h) = \textbf{Algorithm1}(\mathcal{V}_h, \mathcal{E}_h, \mathbf{A}_h, metric_{topK})$       $\triangleright$ Generate LR-HR graph pair
19:     $data$.append$((\mathcal{G}_l, \mathcal{G}_h))$
20: **end for**
21: **return** $data$

---

**Algorithm 3** Data simulation using Barabási-Albert (BA) Model

---

1: **Input:** $N, n, m_{min}, m_{max}, metric_{topK}$
2: **Output:** $data$              $\triangleright$ List of (LR, HR) graph pairs
3: $data \leftarrow []$           $\triangleright$ Initialize an empty list to store graph pairs
4: **for** $l \leftarrow 1$ **to** $N$ **do**
5:     $m \sim \mathcal{U}(m_{min}, m_{max})$        $\triangleright$ Initialize the number of edges from new nodes
6:     $\mathcal{G}_h = BA(n, m)$            $\triangleright$ Create HR graph
7:     $(\mathcal{G}_l, \mathcal{G}_h) = \textbf{Algorithm1}(\mathcal{V}_h, \mathcal{E}_h, \mathbf{A}_h, metric_{topK})$       $\triangleright$ Generate LR-HR graph pair
8: **end for**
9: **return** $data$

---

### C.2.2 LR GRAPH GENERATION

To create LR graphs from the HR graphs, we use the TopK pooling technique (Cangea et al., 2018). For this, we calculate a node metric $metric_{topK}$ for our HR nodes and sort them in decreasing order of this metric. After this, we retain the top $K$ nodes and the connections between to generate the corresponding LR graph. In our experiments, we use four different topological metrics as $metric_{topK}$:

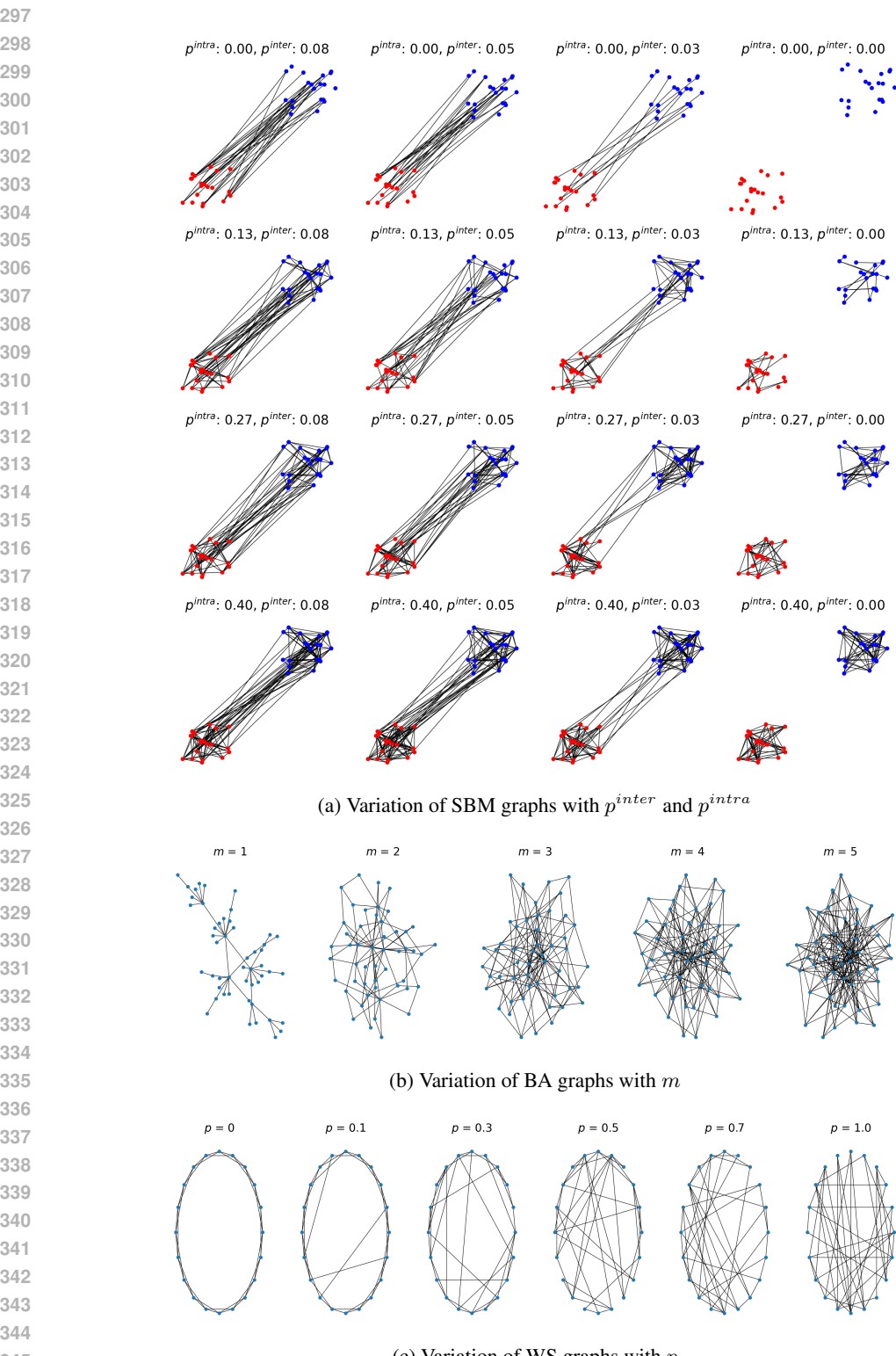

(a) Variation of SBM graphs with $p^{inter}$ and $p^{intra}$

(b) Variation of BA graphs with $m$

(c) Variation of WS graphs with $p$

Figure 8: Sample graphs from different data generation processes and their variation with input parameters

---

**Algorithm 4** Data simulation using Watts Strogatz (WS) Model

---

1: **Input:** $N, n, k_{min}, k_{max}, p_{min}, p_{max}, metric_{topK}$
2: **Output:** $data$         ▷ List of (LR, HR) graph pairs
3: $data \leftarrow []$        ▷ Initialize an empty list to store graph pairs
4: **for** $l \leftarrow 1$ **to** $N$ **do**
5:     $k \sim \mathcal{U}(k_{min}, k_{max})$       ▷ Initialize the number of nearest neighbors
6:     $p \sim \mathcal{U}(p_{min}, p_{max})$        ▷ Initialize the rewiring probability
7:     $\mathcal{G}_h = WS(n, k, p)$         ▷ Create HR graph
8:     $\mathcal{V}_h, \mathcal{E}_h, \mathbf{A}_h = \mathcal{G}_h = WS(n, k, p)$     ▷ Create HR graph structure
9:     $(\mathcal{G}_l, \mathcal{G}_h) = \textbf{Algorithm1}(\mathcal{V}_h, \mathcal{E}_h, \mathbf{A}_h, metric_{topK})$   ▷ Generate LR-HR graph pair
10:    $data.\text{append}((\mathcal{G}_l, \mathcal{G}_h))$
11: **end for**
12: **return** $data$

---

(1) Node Degree Centrality (*Degree*), (2) Betweenness Centrality (*Betweenness*), (3) Clustering Co-efficient (*Clustering*), (4) Participation Coefficient (*Participation*). These metrics were selected as they give rise to different HR-LR graph relationships (see Figure 9), allowing us to cover a diverse set of real-world scenarios.

### C.2.3 NODE FEATURE GENERATION

To generate the initial node feature matrix $\mathbf{X}$ for our graphs, we use the *Node2Vec* model (Grover & Leskovec, 2016). The *Node2Vec* model combines random walks with the *Word2Vec* algorithm (Mikolov, 2013) to generate node embeddings. Specifically, it generates a set of random walks following both the breadth first search (BFS) approach (Bundy & Wallen, 1984) and the depth first search (DFS) approach (Tarjan, 1972). BFS explore nodes closer to the current node, capturing local properties while DFS generates walks exploring nodes further away, capturing global graph properties. Then, it treats each walk as a sentence and applies the *Word2Vec* model to generate our final node feature vectors.

### C.2.4 EDGE WEIGHT GENERATION

To facilitate edge weight prediction, we also generate the edge weighted matrix $\mathbf{E} \in \mathbb{R}^{n \times n}$ for our graphs. Each edge weight is computed as the Pearson correlation coefficient between incident node feature vectors:

$$\mathbf{E}_{ij} = \frac{Cov(\mathbf{X}_i, \mathbf{X}_j)}{\sigma(\mathbf{X}_i)\sigma(\mathbf{X}_j)} \tag{48}$$

where $\mathbf{X}_i$ and $\mathbf{X}_j$ are the feature vectors of incident nodes $i$ and $j$, respectively.

### C.3 BRAIN GRAPH DATASET

We use the publicly available Southwest University Longitudinal Imaging Multimodal (SLIM) dataset (Liu et al., 2017), which provides a collection of structural, diffusion, and resting-state functional magnetic resonance imaging (fMRI) data for 167 subjects. In addition to neuroimaging data, the dataset also contains behavioral data, offering a multifaceted view of brain structure and function. This dataset is used to generate different brain connectivity matrices for each subject using multi-step, complex, and computationally expensive pre-processing pipelines. The brain connectivity matrices vary widely in resolution and each represents a specific type of brain connectome, such as the structural connectome which models anatomical connectivity or the functional connectome which models neural activity between brain regions. Depending on how we parcellate the brain into regions of interests (ROIs) or nodes, we obtain functional connectomes of different resolution. Moreover, the brain connectivity matrices for these connectomes encode neural activity correlation between different ROIs.

For our experiments, we generate LR-HR brain graph pairs using two such functional connectomes: Dosenbach parcellated connectomes (Dosenbach et al., 2010) with 160 ROIs as the LR graphs and Shen parcellated connectomes (Shen et al., 2013) with 268 ROIs as the HR graphs. Figure 10 illustrate some sample LR-HR connectivity matrices for these connectomes, highlighting the topological

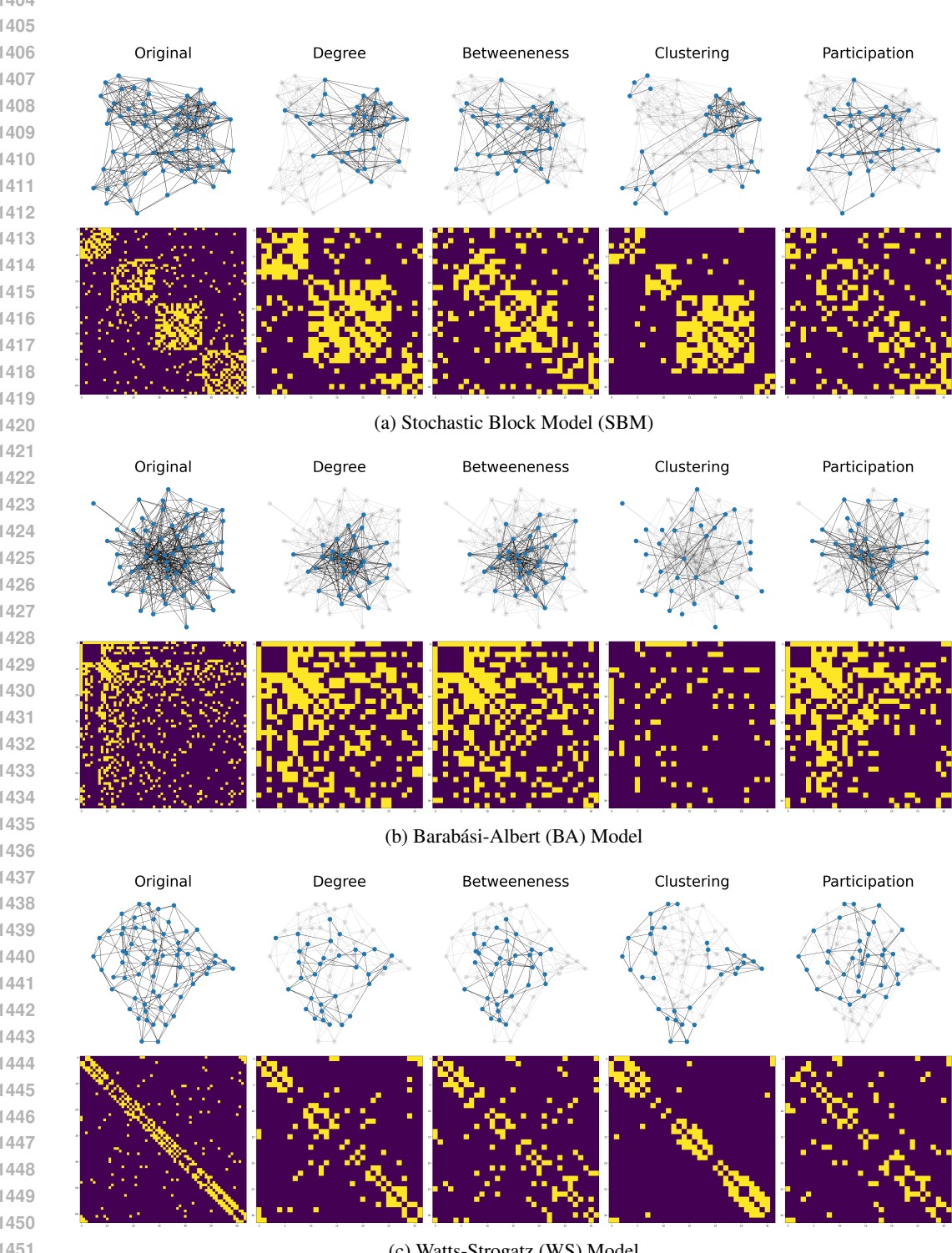

Figure 9: Variation of LR-HR graph pairs across different $metric_{topK}$: (1)*Degree*, (2)*Betweenness*, (3)*Clustering*, (4)*Participation*. Column 'Original' refers to a sample HR graph while others represent the corresponding LR graph. Also, top row in each subfigure show the graph structures while bottom row show the adjacency matrices.

diversity of our dataset. These connectivity matrices form the weighted adjacency matrices for our for LR and HR graphs, denoted by $\mathbf{A}_l$ and $\mathbf{A}_h$, respectively. Following the convention from previous work (Mhiri et al., 2021), we initialize our LR and HR node feature matrices as $\mathbf{X}_l = \mathbf{A}_l$ and $\mathbf{X}_h = \mathbf{A}_h$.

## LR Connectivity Matrices

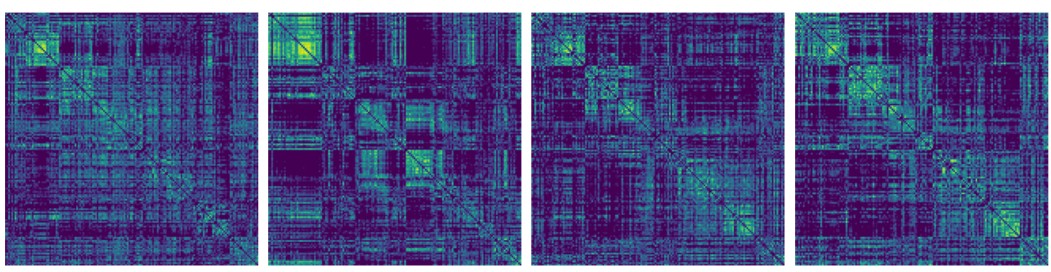

## HR Connectivity Matrices

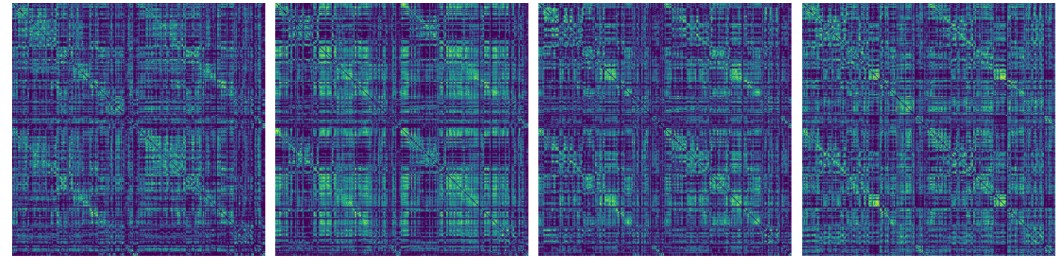

Figure 10: Representative samples of connectivity matrices for our LR-HR brain graphs

# D    COMPARISON MODELS

## D.1    PHYSICS-INSPIRED DUMMY DATASET

To evaluate our proposition, we create four simple models, two each for node and edge representation learning:

- **Node Model**: Inspired by the GIN layer Xu et al. (2018), we create a single layer MPNN that updates node representation and predicts the edges as:

$$\hat{\mathbf{x}}_i = f_{node}(\mathbf{x}_i + \sum_{j \in \mathcal{N}_i} \mathbf{x}_j)$$

$$e_{ij} = \hat{\mathbf{x}}_i \cdot \hat{\mathbf{x}}_j \tag{49}$$

  where, $f_{node}$ is a universal function approximator modeled as a two-layer feed forward network (FFN) Hornik et al. (1989) s.t. $f_{node} : \mathbb{R}^3 \mapsto \mathbb{R}^{16} \mapsto \mathbb{R}^{16}$.

- **Node Large Model**: Same as the above model but with a larger three-layer FFN as the universal function approximator $f_{node\_large} : \mathbb{R}^3 \mapsto \mathbb{R}^{16} \mapsto \mathbb{R}^{16} \mapsto \mathbb{R}^1$. Although $f_{node\_large}$ has larger capacity, it projects node features to a single value in the last layer and thus may struggle for equations that require dot product between larger feature vectors.

- **Edge Model**: Uses simple edge based computations that only depend on adjacent node features as:

$$\mathbf{e}_{ij}^0 = [\mathbf{x}_i || \mathbf{x}_j]$$

$$e_{ij} = f_{edge}(\mathbf{e}_{ij}^0) \tag{50}$$

  where, $||$ is the concatenation operator and $f_{edge}$ is a three-layer FFN $f_{edge} : \mathbb{R}^6 \mapsto \mathbb{R}^{16} \mapsto \mathbb{R}^{16} \mapsto \mathbb{R}^1$.

- **Dual Edge Model**: Involves message passing between the edges using our dual graph formulation. Formally:

$$\mathbf{e}_{ij}^0 = [\mathbf{x}_i || \mathbf{x}_j]$$

$$e_{ij} = f_{edge\_dual}(\mathbf{e}_{ij}^0 + \sum_{k \in \mathcal{N}_i} \mathbf{e}_{ik} + \sum_{l \in \mathcal{N}_j} \mathbf{e}_{lj} + \sum_{s \in \mathcal{N}_i} \mathbf{e}_{si} + \sum_{r \in \mathcal{N}_j} \mathbf{e}_{jr}) \quad (51)$$

where, $f_{edge\_dual}$ is a three-layer FFN network $f_{edge\_dual} : \mathbb{R}^6 \mapsto \mathbb{R}^{16} \mapsto \mathbb{R}^{16} \mapsto \mathbb{R}^1$.

## D.2 TRADITIONAL GRAPH SIMULATION DATASET

To benchmark our frameworks against the simulated datasets, we create two sets of ablated models: (1) Seven models for the super-resolution operator $\mathcal{S}$; (2) Seven models supplementing our models from set one with the dual graph operator $\mathcal{D}$. Below, we discuss the nomenclature used for the models in the first set:

- **LA**: $\mathcal{S}$ with the linear algebraic method
- **Bi-LC**: $\mathcal{S}$ with the bipartite linear combination method
- **Bi-LC$_{fixed}$**: $\mathcal{S}$ with the bipartite linear combination method and node refinement using fixed computation domain
- **Bi-LC$_{learned}$**: $\mathcal{S}$ with the bipartite linear combination method and node refinement using learned computation domain
- **Bi-MP**: $\mathcal{S}$ with the bipartite message passing method
- **Bi-MP$_{fixed}$**: $\mathcal{S}$ with the bipartite message passing method and node refinement using fixed computation domain
- **Bi-MP$_{learned}$**: $\mathcal{S}$ with the bipartite message passing method and node refinement using learned computation domain

As the models in the second set simply use the dual graph operator $\mathcal{D}$ as an additional component, we define the corresponding models as: **Dual LA**, **Dual Bi-LC**, **Dual Bi-LC$_{fixed}$**, **Dual Bi-LC$_{learned}$**, **Dual Bi-MP**, **Dual Bi-MP$_{fixed}$**, and **Dual Bi-MP$_{learned}$**.

## D.3 BRAIN GRAPH DATASET

To thoroughly evaluate our frameworks, we use the fourteen ablated models from section D.2 and benchmark them against an adapted version of the current state of the art GNN model for graph super-resolution and a newly created baseline:

- **IMAN$_{adapted}$**: IMANGraphNet (Mhiri et al., 2021) is the current state of the art GNN model for graph super-resolution. However, it uses computationally expensive NNConv layers (Simonovsky & Komodakis, 2017) and results in 'Out-of-Memory' error on our dataset. Therefore, we create an adapted version of this model which linearly projects the node feature matrix $\mathbf{X}_l$ to a lower dimensional space before feeding it to the NNConv layers. Moreover, to maintain dimensional consistency with the original model, we apply another linear projection to map the outputs back to the higher dimensional space.
- **Autoencoder**: Inspired by the iterative up-and-down sampling methods in image super-resolution (Haris et al., 2018), we propose an autoencoder model to capture the mutual dependency of LR and HR graphs. Both encoder and decoder use the same GNN architecture as our **LA** model but with the mappings reversed s.t. the encoder predicts HR graph from the LR graph while the decoder maps the predicted HR graph back to the original LR graph. Finally, the model is trained using the sum of reconstruction loss for both HR and LR graphs.

# E EXPERIMENTAL SET-UP

## E.1 PHYSICS-INSPIRED DUMMY DATASET

We conduct two sets of experiments, covering eight different scenarios: (1) three experiments with fixed edge function **E1** and varying datasets **D1**, **D2**, and **D3** (2) five experiments for fixed dataset

**D3** and varying edge functions **E1**, **E2**, **E3**, **E4**, and **E5**. For the first set of experiments with **E1**, we fix $G = 100.0$, $G = 1.0$, and $G = 1.0$ for **D1**, **D2**, and **D3** datasets, respectively. These $G$ values were selected empirically to ensure that the resulting edge values are not vanishingly small or explodingly large. For the second set of experiments with **D3**, we fix $G = 1.0$ for **E1** and $A = 10$ and $B = -7$ for **E2**.

For each experiment, we randomly generate three datasets: train, val, and test. We use the train dataset to train the models, val dataset to check for early stopping, and test dataset to report final performance on the best model. All models are trained at least until a given number of warm up epochs. Thereafter, we monitor validation loss and cease training early if it doesn't improve for a given number of epochs, called *patience*. Moreover, we repeat each experiment 15 times to account for variation in the data generation process. All models are trained with MSE loss as it provides a smoother loss landscape which is preferable for our simplistic setting. Finally, Table 5 provides common hyper-parameters used across all experiments.

Table 5: Hyper-parameters for experiments with physics-inspired dummy dataset

| Hyper-parameter type | Hyper-parameter | Value |
|---|---|---|
| Data Generation | *Number of nodes* | 16 |
| | *Number of train samples* | 128 |
| | *Number of val samples* | 32 |
| | *Number of test samples* | 32 |
| | *Connection threshold, $t$* | 0.3 |
| Model training | *Batch size* | 16 |
| | *Learning rate* | 0.001 |
| | *Maximum number of epochs* | 300 |
| | *Number of warmup epochs* | 10 |
| | *Patience* | 15 |

### E.2 TRADITIONAL GRAPH SIMULATION DATASET

Our objective is to predict the HR edge features $\mathbf{E}_h$ from the LR edge features $\mathbf{E}_l$ under twelve different scenarios covering three graph topology and four metrics for TopK pooling. We evaluate each scenario using 3-fold cross validation. For each fold, we split the dataset into train, val, and test. Similar to section E.1, we use train dataset for model training, val dataset to determine early stopping, and test dataset to report performance for that fold. We average this performance across all folds to report final model performance. All models are trained with MAE loss between predicted and true $\mathbf{E}_h$. Table 6 gives the hyper-parameters used for our experiments.

### E.3 BRAIN GRAPH DATASET

Our objective is to predict the HR adjacency matrix $\mathbf{A}_h$ from the LR adjacency matrix $\mathbf{A}_l$ and analyze the performance across sixteen different models. We use the same experimental setting as section E.2 but with some minor changes: (1) We perform categorical search on learning rate and select the learning rate with the best performance for each model from [0.01, 0.005, 0.001]. (2) We use the hyper-parameters given in Table 7 for model training and Graph Transformer Block (GTB).

Along with the MAE between the true and predicted $\mathbf{A}_h$, we also measure the MAE between seven topological measures: Betweenness Centrality (*Betweenness*), Closeness Centrality (*Closeness*), Eigenvector Centrality (*Eigenvector*), Node Degree Centrality (*Degree*), Participation Centrality (*Participation*), Clustering Coefficient (*Clustering*), and Small Worldness (*Small Worldness*). Each one of these measures capture a different topological aspect of the connectome.

Node degree centrality measures the number of incident connections to a given node and serves as an indirect measure of network resilience (Achard et al., 2006). Betweenness centrality measures the fraction of shortest paths between all node pairs that pass through a given node and is useful for detecting bridge nodes between disparate regions (Rubinov & Sporns, 2010). Closeness centrality quantifies the mean distance between a given node and the rest of the network, indicating the speed

Table 6: Hyper-parameters for experiments with simulated datasets

| Hyper-parameter type | Hyper-parameter | Value |
|---|---|---|
| Model training | *Batch size* | 16 |
| | *Learning rate* | 0.001 |
| | *Maximum number of epochs* | 150 |
| | *Number of warmup epochs* | 15 |
| | *Patience* | 5 |
| GTB parameters | *Number of hidden dims* | 16 |
| | *Number of attention heads* | 4 |
| | *Dropout* | 0.2 |
| Data Generation | *Number of samples* | 128 |
| | *Number of HR nodes, $n$* | 64 |
| | *Number of LR nodes, $K$* | 32 |
| SBM parameters | *Minimum number of clusters, $c_{min}$* | 2 |
| | *Maximum number of clusters, $c_{max}$* | 5 |
| | *Minimum intra connection probability, $p_{min}^{intra}$* | 0.50 |
| | *Maximum intra connection probability, $p_{max}^{intra}$* | 0.60 |
| | *Minimum inter connection probability, $p_{min}^{inter}$* | 0.01 |
| | *Maximum inter connection probability, $p_{max}^{inter}$* | 0.10 |
| BA parameters | *Minimum number of edges, $m_{min}$* | 4 |
| | *Maximum number of edges, $m_{max}$* | 8 |
| WS parameters | *Minimum number of nearest neighbors, $k_{min}$* | 4 |
| | *Maximum number of nearest neighbors , $k_{max}$* | 8 |
| | *Minimum rewiring probability, $p_{min}$* | 0.2 |
| | *Maximum rewiring probability, $p_{max}$* | 0.5 |
| Node2Vec parameters | *Node feature dimension* | 8 |
| | *Length of HR random walks* | 51 |
| | *Length of LR random walks* | 26 |
| | *Number of random walks* | 100 |

Table 7: Hyper-parameters for experiments with the brain graph dataset.

| Hyper-parameter type | Hyper-parameter | Value |
|---|---|---|
| Model training | *Batch size* | 16 |
| | *Maximum number of epochs* | 300 |
| | *Number of warmup epochs* | 30 |
| | *Patience* | 7 |
| GTB parameters | *Number of hidden dims* | 32 |
| | *Number of attention heads* | 4 |
| | *Dropout* | 0.2 |
| Dataset parameters | *Number of LR nodes* | 160 |
| | *Number of HR nodes* | 268 |

of communication within the network. Eigenvector centrality assess the number of connections to a given node, weighted by the centrality of its neighbors, and evaluates hierarchical influence (Lorenzini et al., 2023). Participation Coefficient and Clustering Coefficient measures modularity in the network. The Participation Coefficient measures the diversity of intermodular interconnections of individual nodes, while the Clustering Coefficient assesses the presence of cliques or clusters. These metrics are important for evaluating brain network segregation and information processing within specialized brain subsystems (Gamboa et al., 2014). Finally, Small-worldness is defined by the ratio between the characteristic path length and mean clustering coefficient (normalized by the corresponding values calculated on random graphs). It supports both segregated/specialized and distributed/integrated information processing (Watts & Strogatz, 1998).

# F  RESULTS

## F.1  PHYSICS-INSPIRED DUMMY DATASET

From table 8 and 9, we observe that the performances are in line with expectation. In the first set of experiments, we fix the edge function to *E1* and vary the datasets. *E1* represents the inverse square law and should be easy to model using node based models when $r_{ij}$ is constant. Consequently, both node based models outperform edge based models on *D1*. However, *E1* is challenging to model using dot product when it solely relies on $1/r_{ij}^2$. As a result, both edge based models outperform the node based ones on *D2*. For *D3*, the numerator seems to compensate for the error from denominator, allowing node based models to achieve performance that is comparable to the edge based models.

Table 8: Test MAE between true and predicted edge value for *E1* (inverse square law) across *D1* (grid graph with random masses), *D2* (random graph with uniform mass), and *D3* (random graph with random masses) datasets. **Bold underline** and **bold** represent the best and second best model across each row or dataset.

| Dataset | Node | Node Large | Edge | Edge Dual |
|---|---|---|---|---|
| *D1* | $\mathbf{0.869 \pm 0.032}$ | $\mathbf{1.136 \pm 0.899}$ | $2.371 \pm 2.087$ | $1.565 \pm 1.317$ |
| *D2* | $41.176 \pm 25.567$ | $39.525 \pm 28.190$ | $\mathbf{33.266 \pm 16.387}$ | $\mathbf{38.221 \pm 23.984}$ |
| *D3* | $13.499 \pm 9.805$ | $\mathbf{9.012 \pm 5.058}$ | $\underline{\mathbf{8.696 \pm 5.444}}$ | $10.873 \pm 5.928$ |

In the second set of experiments, we use *D3* as our dataset and vary the edge function. For both *E1* and *E2*, the compensatory effect between numerator and denominator terms takes place and the node based models perform on par with their edge based counterparts. However, this compensatory effect is absent in *E3* leading to both node based models struggling and performing poorly. For *E4* and *E5*, we anticipates the models to utilize higher dimensional sparse representations. Therefore, models with the FFN projecting to a single value in the last layer tend to suffer. However, our dual edge model is able to outperform the other edge model possibly due to its larger capacity and corrections to the final edge value via message passing from other edges. For the other edge types, this larger capacity and message passing operation seemed unhelpful and even counterproductive possibly due to small dataset size and relatively simpler edge functions.

Finally, we would like to highlight some caveats in our experiment design. First, we observe a high variance between runs and significant outliers (see Figure 11). This may occur since our data generation is not controlled and could lead to a very large edge value when two particles are generated closely. As our training objective is to minimize the MSE loss, this creates a bias in the model and may lead to incorrect estimation of model performance. We tried to correct for this phenomena by averaging performance across a larger number of runs. Second, the individual models have not been tuned for best performance and the experiments only act as a proof of concept to highlight the general trend.

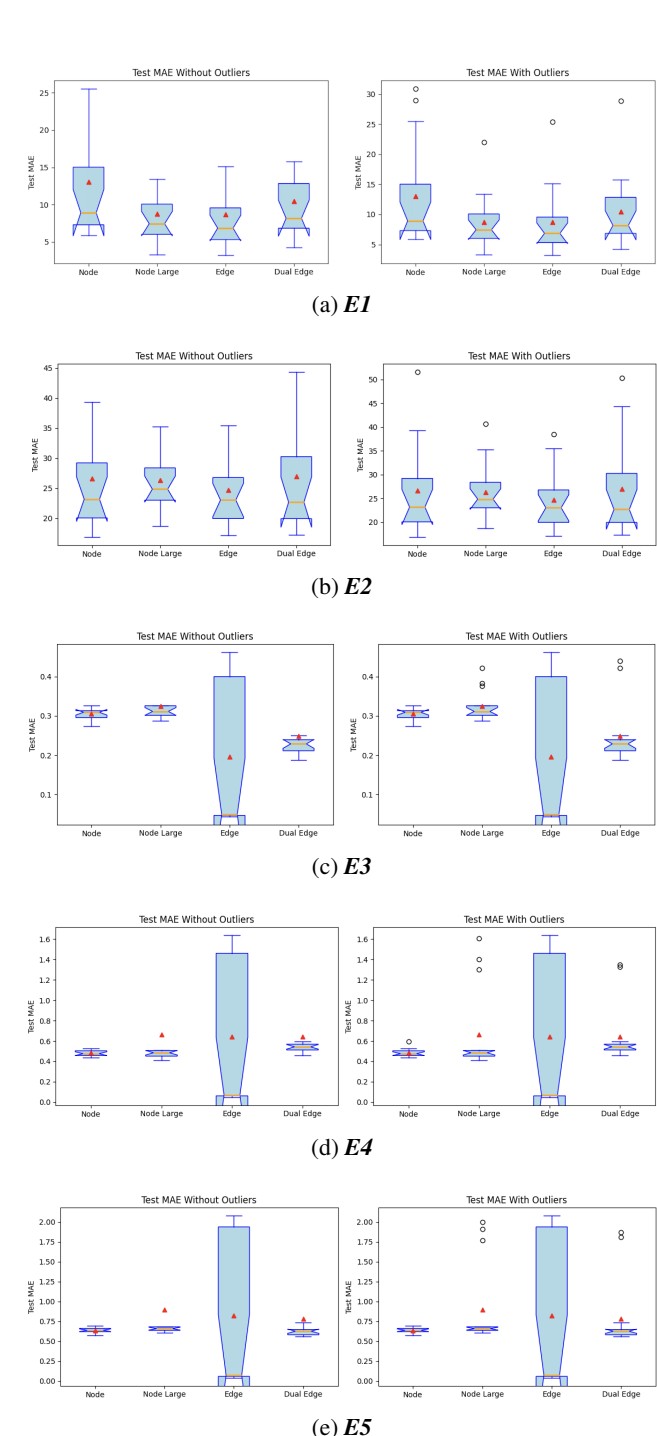

Figure 11: Performance variation across 15 runs for the **D3** dataset. Random and uncontrolled data generation causes high variance and outliers. However, as the experiments are run a large number of times, the average performance (represented by ▲) is expected to be representative of the true model performance.

Table 9: Test MAE between true and predicted edge value for **D3** (random graph with random masses) dataset across **E1** (inverse square law), **E2** (asymmetric rational), **E3** (symmetric quadratic), **E4** (symmetric polynomial), and **E5** (asymmetric quadratic) edge functions. **Bold underline** and **bold** represent the best and second best models across each edge function.

| Edge Function | Node | Node Large | Edge | Edge Dual |
|---|---|---|---|---|
| **E1** | $13.499 \pm 9.805$ | $\mathbf{9.012 \pm 5.058}$ | $\underline{\mathbf{8.696 \pm 5.444}}$ | $10.873 \pm 5.928$ |
| **E2** | $26.611 \pm 9.176$ | $\mathbf{26.304 \pm 5.800}$ | $\underline{\mathbf{24.702 \pm 6.480}}$ | $26.991 \pm 10.280$ |
| **E3** | $0.305 \pm 0.014$ | $0.325 \pm 0.037$ | $\underline{\mathbf{0.196 \pm 0.182}}$ | $\mathbf{0.249 \pm 0.073}$ |
| **E4** | $\mathbf{0.485 \pm 0.039}$ | $0.663 \pm 0.391$ | $0.640 \pm 0.710$ | $\mathbf{0.639 \pm 0.275}$ |
| **E5** | $\underline{\mathbf{0.637 \pm 0.036}}$ | $0.898 \pm 0.500$ | $0.821 \pm 0.934$ | $\mathbf{0.779 \pm 0.419}$ |

## F.2 TRADITIONAL GRAPH SIMULATION DATASET

We observe that the simulation scenarios are deceptively simple. For example, from figure 9a, notice that that there are 4 clusters in the HR graph yet only 3 in the LR graph for *Degree*. Such a scenario would be challenging for our models to learn since this requires predicting HR edges for the missing cluster with barely any nodes from that cluster in the LR graph. Still, we observe that our bipartite graph formulation outperforms the linear algebraic method across all experiments.

From Table 10, 11, and 12, we observe that bipartite message passing performs better than bipartite linear combination across most of the scenarios. Bi-MP models clearly outperforms Bi-LC models for BA and WS dataset while the performance is very close for the SBM dataset. This could be possibly because linear combination provides a highly flexible approach that is useful for predicting edges from the missing clusters while message passing doesn't add much utility if no nodes from the missing cluster are present. For WS and BA datasets, observe from figure 9b and 9c that the HR graph looks like an extrapolated version of each LR graph and thus, message passing may be helpful to learn the underlying relationship between nodes. We also do not observe performance gain from using our dual graph operator $\mathcal{D}$. We suspect this to follow from the previous section where we observed that edge based message passing does not provide additional utility for small graphs where node based models may suffice.

Table 10: Test MAE for $\mathbf{E}_h$ on four SBM datasets. Columns refer to $metric_{topK}$ datasets partitioned between models with and without $\mathcal{D}$. In each column, colors give the top 3 models while **bold + underline** and **bold** gives the best and second best model for each partition.

| Model | Degree | Betweenness | Clustering | Participation |
|---|---|---|---|---|
| LA | $2.841 \pm 0.123$ | $2.712 \pm 0.204$ | $2.591 \pm 0.087$ | $2.784 \pm 0.062$ |
| Bi-LC | $\mathbf{2.495 \pm 0.109}$ | $\mathbf{2.603 \pm 0.021}$ | $\mathbf{2.463 \pm 0.061}$ | $\mathbf{2.570 \pm 0.061}$ |
| Bi-LC$_{fixed}$ | $2.518 \pm 0.119$ | $2.626 \pm 0.015$ | $2.574 \pm 0.091$ | $2.578 \pm 0.076$ |
| Bi-LC$_{learned}$ | $2.595 \pm 0.139$ | $2.678 \pm 0.049$ | $2.574 \pm 0.051$ | $2.592 \pm 0.072$ |
| Bi-MP | $2.548 \pm 0.103$ | $2.685 \pm 0.030$ | $2.494 \pm 0.057$ | $2.592 \pm 0.083$ |
| Bi-MP$_{fixed}$ | $\mathbf{2.511 \pm 0.117}$ | $\mathbf{2.594 \pm 0.009}$ | $\mathbf{2.463 \pm 0.039}$ | $\mathbf{2.572 \pm 0.103}$ |
| Bi-MP$_{learned}$ | $2.523 \pm 0.123$ | $2.659 \pm 0.066$ | $2.553 \pm 0.084$ | $2.691 \pm 0.090$ |
| Dual LA | $3.384 \pm 0.962$ | $3.350 \pm 1.059$ | $2.600 \pm 0.044$ | $3.437 \pm 1.214$ |
| Dual Bi-LC | $\mathbf{2.558 \pm 0.136}$ | $\mathbf{2.637 \pm 0.028}$ | $\mathbf{2.511 \pm 0.063}$ | $2.994 \pm 0.721$ |
| Dual Bi-LC$_{fixed}$ | $3.286 \pm 0.565$ | $2.887 \pm 0.307$ | $2.615 \pm 0.166$ | $2.916 \pm 0.132$ |
| Dual Bi-LC$_{learned}$ | $2.589 \pm 0.150$ | $3.404 \pm 0.977$ | $3.066 \pm 0.600$ | $2.750 \pm 0.129$ |
| Dual Bi-MP | $2.562 \pm 0.160$ | $2.660 \pm 0.069$ | $2.524 \pm 0.016$ | $\mathbf{2.539 \pm 0.044}$ |
| Dual Bi-MP$_{fixed}$ | $2.601 \pm 0.119$ | $2.668 \pm 0.032$ | $2.676 \pm 0.143$ | $\mathbf{2.625 \pm 0.019}$ |
| Dual Bi-MP$_{learned}$ | $\underline{\mathbf{2.543 \pm 0.093}}$ | $\underline{\mathbf{2.629 \pm 0.017}}$ | $\mathbf{2.493 \pm 0.088}$ | $2.694 \pm 0.220$ |

Table 11: Test MAE for $\mathbf{E}_h$ on four BA datasets. Columns refer to $metric_{topK}$ datasets partitioned between models with and without $\mathcal{D}$. In each column, colors give the top 3 models while **bold + underline** and **bold** gives the best and second best model for each partition.

| Model | Degree | Betweenness | Clustering | Participation |
|---|---|---|---|---|
| LA | $1.761 \pm 0.107$ | $2.162 \pm 0.364$ | $1.813 \pm 0.075$ | $1.747 \pm 0.055$ |
| Bi-LC | $1.749 \pm 0.033$ | $1.789 \pm 0.038$ | $1.787 \pm 0.054$ | $1.752 \pm 0.028$ |
| Bi-LC$_{fixed}$ | $1.738 \pm 0.018$ | $\mathbf{1.767 \pm 0.022}$ | $1.814 \pm 0.054$ | $\mathbf{1.728 \pm 0.037}$ |
| Bi-LC$_{learned}$ | $1.745 \pm 0.018$ | $1.775 \pm 0.058$ | $1.833 \pm 0.028$ | $1.743 \pm 0.044$ |
| Bi-MP | $\mathbf{1.721 \pm 0.026}$ | $\mathbf{1.750 \pm 0.038}$ | $\mathbf{1.753 \pm 0.065}$ | $\mathbf{1.726 \pm 0.080}$ |
| Bi-MP$_{fixed}$ | $\underline{\mathbf{1.705 \pm 0.019}}$ | $1.795 \pm 0.040$ | $\mathbf{1.748 \pm 0.093}$ | $1.758 \pm 0.029$ |
| Bi-MP$_{learned}$ | $1.789 \pm 0.041$ | $1.780 \pm 0.039$ | $1.763 \pm 0.052$ | $1.761 \pm 0.007$ |
| Dual LA | $1.845 \pm 0.047$ | $1.894 \pm 0.050$ | $1.861 \pm 0.096$ | $1.800 \pm 0.031$ |
| Dual Bi-LC | $\mathbf{1.775 \pm 0.035}$ | $1.802 \pm 0.051$ | $\underline{\mathbf{1.775 \pm 0.083}}$ | $\mathbf{1.732 \pm 0.028}$ |
| Dual Bi-LC$_{fixed}$ | $1.984 \pm 0.222$ | $2.446 \pm 0.891$ | $1.966 \pm 0.084$ | $2.659 \pm 1.209$ |
| Dual Bi-LC$_{learned}$ | $2.459 \pm 0.880$ | $2.604 \pm 0.712$ | $2.416 \pm 0.905$ | $3.030 \pm 0.924$ |
| Dual Bi-MP | $1.824 \pm 0.018$ | $1.974 \pm 0.091$ | $1.869 \pm 0.024$ | $1.863 \pm 0.130$ |
| Dual Bi-MP$_{fixed}$ | $1.931 \pm 0.087$ | $\mathbf{1.797 \pm 0.073}$ | $1.888 \pm 0.098$ | $1.803 \pm 0.088$ |
| Dual Bi-MP$_{learned}$ | $\underline{\mathbf{1.721 \pm 0.041}}$ | $\underline{\mathbf{1.784 \pm 0.009}}$ | $\mathbf{1.809 \pm 0.092}$ | $\underline{\mathbf{1.790 \pm 0.144}}$ |

Table 12: Test MAE for $\mathbf{E}_h$ on four WS datasets. Columns refer to $metric_{topK}$ datasets partitioned between models with and without $\mathcal{D}$. In each column, colors give the top 3 models while **bold + underline** and **bold** gives the best and second best model for each partition.

| Model | Degree | Betweenness | Clustering | Participation |
|---|---|---|---|---|
| LA | $2.179 \pm 0.132$ | $2.070 \pm 0.061$ | $2.128 \pm 0.053$ | $2.104 \pm 0.100$ |
| Bi-LC | $\underline{\mathbf{1.989 \pm 0.006}}$ | $2.007 \pm 0.012$ | $2.002 \pm 0.031$ | $2.022 \pm 0.027$ |
| Bi-LC$_{fixed}$ | $2.035 \pm 0.035$ | $2.058 \pm 0.022$ | $2.034 \pm 0.015$ | $2.075 \pm 0.066$ |
| Bi-LC$_{learned}$ | $2.012 \pm 0.020$ | $2.043 \pm 0.043$ | $2.027 \pm 0.067$ | $2.090 \pm 0.046$ |
| Bi-MP | $\mathbf{1.998 \pm 0.020}$ | $\mathbf{2.002 \pm 0.005}$ | $2.016 \pm 0.056$ | $\mathbf{2.013 \pm 0.017}$ |
| Bi-MP$_{fixed}$ | $2.003 \pm 0.014$ | $\underline{\mathbf{1.996 \pm 0.010}}$ | $\mathbf{1.998 \pm 0.027}$ | $2.019 \pm 0.028$ |
| Bi-MP$_{learned}$ | $2.060 \pm 0.083$ | $2.028 \pm 0.035$ | $\underline{\mathbf{1.994 \pm 0.030}}$ | $\underline{\mathbf{2.010 \pm 0.025}}$ |
| Dual LA | $2.149 \pm 0.011$ | $2.879 \pm 1.197$ | $2.156 \pm 0.027$ | $2.206 \pm 0.085$ |
| Dual Bi-LC | $\mathbf{2.027 \pm 0.040}$ | $\mathbf{2.007 \pm 0.016}$ | $\mathbf{2.009 \pm 0.024}$ | $2.422 \pm 0.698$ |
| Dual Bi-LC$_{fixed}$ | $2.615 \pm 0.894$ | $3.025 \pm 1.602$ | $2.466 \pm 0.649$ | $2.320 \pm 0.284$ |
| Dual Bi-LC$_{learned}$ | $2.679 \pm 1.055$ | $2.942 \pm 0.802$ | $2.140 \pm 1.846$ | $2.303 \pm 0.199$ |
| Dual Bi-MP | $2.450 \pm 0.598$ | $\mathbf{2.039 \pm 0.038}$ | $\mathbf{2.083 \pm 0.122}$ | $\mathbf{2.097 \pm 0.049}$ |
| Dual Bi-MP$_{fixed}$ | $2.270 \pm 0.381$ | $2.394 \pm 0.414$ | $2.432 \pm 0.683$ | $2.417 \pm 0.622$ |
| Dual Bi-MP$_{learned}$ | $\underline{\mathbf{2.022 \pm 0.048}}$ | $2.970 \pm 1.666$ | $2.427 \pm 0.710$ | $\underline{\mathbf{2.041 \pm 0.049}}$ |

Finally, we highlight some experimental caveats. First, we perform our experiments on small graphs and small data regime. While small graphs are found plenty in the graph learning tasks, neural networks generally struggle with small datasets and are prone to overfitting. This could be circumvented by performing scaling analysis for our frameworks but this is beyond the scope of this work. Second, topK pooling uses traditional metric to create a relationship between LR and HR nodes. This may not be reflective of real-world graphs that encode more complex non-hierarchial relationships.

### F.3 Brain Graph Dataset

We report the performance across all eight evaluation measures in Table 13. We observe that our dual graph formulation outperforms other methods, especially across the topological measures. It beats the IMAN$_{adapted}$ and Autoencoder by a wide margin on these measures. For our bipartite graph formulation, we observe that message passing performs better than linear combination in the absence of the dual graph operator $\mathcal{D}$ but the performance difference diminishes on supplmenting the models with $\mathcal{D}$. This could be possibly because our dual graph formulation provides a powerful and robust framework to refine the initially learned edge features from $\mathcal{S}$, uplifting the performance of the the linear combination method. Unfortunately, the bipartite graph formulation does not improve over the linear algebraic method for this specific brain graph dataset.

### F.4 Sensitivity Analysis

Finally, we also perform an in-depth sensitivity analysis for the random initialization strategy introduced for our bipartite message passing framework in section 3. Recall that this strategy involves initializing an HR node feature matrix with values randomly sampled from $\mathcal{U}(0, 1)$. To analyze how sensitive our model performance is to this initialization, we re-run our experiments 15 times for the six models based on bipartite message passing: **Bi-MP**, **Bi-MP**$_{fixed}$, **Bi-MP**$_{learned}$, **Dual Bi-MP**, **Dual Bi-MP**$_{fixed}$, and **Dual Bi-MP**$_{learned}$. These 15 runs measure performance across 5 different random seeds and 3 length scales viz $\mathcal{U}(0, 1), \mathcal{U}(0, 10)$, and $\mathcal{U}(0, 100)$. To measure the sensitivity of our formulation w.r.t. the other models, we introduce a quantitative metric called relative sensitivity $s_{rel}$ as:

$$s_{rel} = \frac{max(\{\sigma_{sm} | s \in scales, m \in models_{Bi-MP}\})}{\sigma_{all\_models}} \tag{52}$$

where, $\sigma_{sm}$ is the standard deviation of the mean MAE loss (averaged across five random seeds) for Bi-MP model $m$ and scale $s$ and $\sigma_{all\_models}$ is the standard deviation of the MAE losses for all sixteen models from section D.3.

Finally, we report the output of our sensitivity analysis in Table 14 and 15. While all bipartite message passing models seem robust against variations in the initialization strategy, we observe that the models without dual graph formulation show a lot more robustness compared to the models with dual graph formulation. This is expected since the dual graph models possess higher capacity and high capacity neural networks generally show less robustness against randomization, especially for small data regime such as ours.

Table 13: Model Performance on Brain Graph Dataset. Each column represents MAE on given evaluation measure. The best and second best models are highlighted by **bold + underline** and **bold** and the colors give relative ordering.

| Model | $\mathbf{A}_h$ $(10^1)$ | Betweenness $(10^4)$ | Closeness $(10^1)$ | Eigenvector $(10^3)$ |
|---|---|---|---|---|
| IMAN$_{adapted}$ | $1.725 \pm 0.074$ | $7.695 \pm 0.159$ | $1.590 \pm 0.028$ | $7.507 \pm 0.096$ |
| AutoEncoder | $1.381 \pm 0.062$ | $7.608 \pm 0.204$ | $1.520 \pm 0.025$ | $7.179 \pm 0.083$ |
| LA | $\underline{\mathbf{1.350 \pm 0.066}}$ | $7.562 \pm 0.152$ | $1.513 \pm 0.033$ | $7.155 \pm 0.124$ |
| Bi-LC | $1.528 \pm 0.021$ | $7.693 \pm 0.159$ | $1.590 \pm 0.028$ | $7.507 \pm 0.096$ |
| Bi-LC$_{fixed}$ | $1.507 \pm 0.051$ | $7.693 \pm 0.159$ | $1.590 \pm 0.028$ | $7.506 \pm 0.096$ |
| Bi-LC$_{learned}$ | $1.523 \pm 0.055$ | $7.693 \pm 0.159$ | $1.590 \pm 0.028$ | $7.506 \pm 0.096$ |
| Bi-MP | $1.455 \pm 0.031$ | $7.658 \pm 0.208$ | $1.578 \pm 0.043$ | $7.453 \pm 0.182$ |
| Bi-MP$_{fixed}$ | $1.428 \pm 0.052$ | $7.588 \pm 0.156$ | $1.551 \pm 0.039$ | $7.325 \pm 0.127$ |
| Bi-MP$_{learned}$ | $1.443 \pm 0.048$ | $7.586 \pm 0.192$ | $1.554 \pm 0.040$ | $7.342 \pm 0.169$ |
| Dual LA | $1.458 \pm 0.153$ | $5.888 \pm 1.914$ | $1.133 \pm 0.442$ | $7.360 \pm 0.957$ |
| Dual Bi-LC | $1.515 \pm 0.293$ | $5.567 \pm 2.235$ | $\mathbf{0.812 \pm 0.123}$ | $6.736 \pm 1.172$ |
| Dual Bi-LC$_{fixed}$ | $1.609 \pm 0.176$ | $\mathbf{5.376 \pm 0.071}$ | $1.030 \pm 0.012$ | $6.560 \pm 0.172$ |
| Dual Bi-LC$_{learned}$ | $1.646 \pm 0.086$ | $7.318 \pm 0.713$ | $1.249 \pm 0.366$ | $7.504 \pm 0.556$ |
| Dual Bi-MP | $1.488 \pm 0.143$ | $\mathbf{5.446 \pm 0.927}$ | $\mathbf{0.939 \pm 0.059}$ | $6.469 \pm 0.370$ |
| Dual Bi-MP$_{fixed}$ | $1.554 \pm 0.185$ | $5.747 \pm 0.848$ | $1.031 \pm 0.147$ | $\mathbf{6.373 \pm 0.411}$ |
| Dual Bi-MP$_{learned}$ | $\mathbf{1.373 \pm 0.039}$ | $5.742 \pm 0.913$ | $1.046 \pm 0.128$ | $\mathbf{6.379 \pm 0.276}$ |

| Model | Degree $(10^0)$ | Participation $(10^1)$ | Clustering $(10^2)$ | Small Worldness $(10^2)$ |
|---|---|---|---|---|
| IMAN$_{adapted}$ | $54.778 \pm 1.170$ | $6.850 \pm 0.091$ | $14.006 \pm 0.318$ | $8.360 \pm 0.243$ |
| AutoEncoder | $51.697 \pm 1.038$ | $5.552 \pm 1.450$ | $14.193 \pm 0.437$ | $8.260 \pm 0.336$ |
| LA | $51.555 \pm 1.458$ | $5.255 \pm 0.883$ | $14.128 \pm 0.286$ | $8.126 \pm 0.289$ |
| Bi-LC | $54.771 \pm 1.170$ | $6.858 \pm 0.173$ | $14.003 \pm 0.318$ | $8.362 \pm 0.240$ |
| Bi-LC$_{fixed}$ | $54.771 \pm 1.170$ | $6.836 \pm 0.096$ | $14.103 \pm 0.318$ | $8.350 \pm 0.244$ |
| Bi-LC$_{learned}$ | $54.771 \pm 1.170$ | $6.822 \pm 0.106$ | $14.103 \pm 0.318$ | $8.358 \pm 0.243$ |
| Bi-MP | $54.341 \pm 1.730$ | $6.410 \pm 0.849$ | $13.956 \pm 0.369$ | $8.331 \pm 0.287$ |
| Bi-MP$_{fixed}$ | $53.324 \pm 1.650$ | $5.090 \pm 0.837$ | $13.916 \pm 0.272$ | $8.254 \pm 0.243$ |
| Bi-MP$_{learned}$ | $53.521 \pm 1.651$ | $5.576 \pm 0.766$ | $13.866 \pm 0.353$ | $8.270 \pm 0.268$ |
| Dual LA | $38.991 \pm 13.900$ | $3.401 \pm 3.172$ | $11.953 \pm 5.235$ | $5.873 \pm 3.221$ |
| Dual Bi-LC | $\mathbf{31.948 \pm 5.635}$ | $\mathbf{1.330 \pm 0.159}$ | $\mathbf{7.779 \pm 2.068}$ | $\mathbf{3.886 \pm 1.847}$ |
| Dual Bi-LC$_{fixed}$ | $37.555 \pm 0.806$ | $\mathbf{1.382 \pm 0.080}$ | $\mathbf{9.718 \pm 0.358}$ | $4.086 \pm 1.036$ |
| Dual Bi-LC$_{learned}$ | $45.300 \pm 10.049$ | $3.615 \pm 2.714$ | $11.874 \pm 2.623$ | $7.188 \pm 1.320$ |
| Dual Bi-MP | $\mathbf{34.298 \pm 2.567}$ | $1.461 \pm 0.204$ | $10.064 \pm 1.623$ | $\mathbf{3.451 \pm 0.696}$ |
| Dual Bi-MP$_{fixed}$ | $37.568 \pm 4.705$ | $1.497 \pm 0.161$ | $10.397 \pm 1.362$ | $4.076 \pm 1.648$ |
| Dual Bi-MP$_{learned}$ | $37.527 \pm 3.782$ | $1.440 \pm 0.233$ | $10.714 \pm 2.245$ | $5.322 \pm 1.068$ |

Table 14: Result of sensitivity analysis for bipartite message passing without $\mathcal{D}$.

| Metric | Scale | Bi-MP | Bi-MP$_{fixed}$ | Bi-MP$_{learned}$ | $s_{rel}$ |
|---|---|---|---|---|---|
| $\mathbf{A}_h$ ($10^1$) | 1 | $1.488 \pm 0.025$ | $1.417 \pm 0.018$ | $1.420 \pm 0.024$ | |
| | 10 | $1.537 \pm 0.012$ | $1.448 \pm 0.029$ | $1.459 \pm 0.013$ | **0.060** |
| | 100 | $1.614 \pm 0.026$ | $1.564 \pm 0.021$ | $1.566 \pm 0.014$ | |
| Betweenness ($10^4$) | 1 | $7.693 \pm 0.000$ | $7.596 \pm 0.028$ | $7.598 \pm 0.045$ | |
| | 10 | $7.611 \pm 0.013$ | $7.589 \pm 0.034$ | $7.599 \pm 0.007$ | **0.018** |
| | 100 | $7.665 \pm 0.022$ | $7.604 \pm 0.027$ | $7.617 \pm 0.037$ | |
| Closeness ($10^1$) | 1 | $1.590 \pm 0.000$ | $1.555 \pm 0.013$ | $1.555 \pm 0.021$ | |
| | 10 | $1.558 \pm 0.007$ | $1.548 \pm 0.014$ | $1.551 \pm 0.010$ | **0.044** |
| | 100 | $1.580 \pm 0.008$ | $1.558 \pm 0.008$ | $1.563 \pm 0.011$ | |
| Eigenvector ($10^3$) | 1 | $7.506 \pm 0.000$ | $7.350 \pm 0.061$ | $7.346 \pm 0.095$ | |
| | 10 | $7.357 \pm 0.032$ | $7.314 \pm 0.064$ | $7.327 \pm 0.044$ | **0.039** |
| | 100 | $7.460 \pm 0.038$ | $7.362 \pm 0.037$ | $7.383 \pm 0.052$ | |
| Degree ($10^0$) | 1 | $54.771 \pm 0.000$ | $53.511 \pm 0.523$ | $53.467 \pm 0.829$ | |
| | 10 | $53.557 \pm 0.299$ | $53.170 \pm 0.569$ | $53.279 \pm 0.456$ | **0.101** |
| | 100 | $54.414 \pm 0.304$ | $53.562 \pm 0.353$ | $53.780 \pm 0.430$ | |
| Participation ($10^1$) | 1 | $6.838 \pm 0.023$ | $5.489 \pm 0.642$ | $5.435 \pm 0.881$ | |
| | 10 | $5.920 \pm 0.423$ | $5.155 \pm 0.717$ | $5.613 \pm 0.356$ | **0.449** |
| | 100 | $6.692 \pm 0.185$ | $5.759 \pm 0.698$ | $6.166 \pm 0.804$ | |
| Clustering ($10^2$) | 1 | $14.003 \pm 0.000$ | $13.907 \pm 0.035$ | $13.911 \pm 0.030$ | |
| | 10 | $13.938 \pm 0.003$ | $13.941 \pm 0.020$ | $13.954 \pm 0.059$ | **0.015** |
| | 100 | $13.974 \pm 0.002$ | $13.934 \pm 0.053$ | $13.924 \pm 0.071$ | |
| Small Worldness ($10^2$) | 1 | $8.360 \pm 0.000$ | $8.270 \pm 0.032$ | $8.278 \pm 0.037$ | |
| | 10 | $8.282 \pm 0.018$ | $8.253 \pm 0.041$ | $8.264 \pm 0.012$ | **0.015** |
| | 100 | $8.333 \pm 0.020$ | $8.267 \pm 0.042$ | $8.289 \pm 0.037$ | |

Table 15: Result of sensitivity analysis for bipartite message passing with $\mathcal{D}$.

| Metric | Scale | Dual Bi-MP | Dual Bi-MP$_{fixed}$ | Dual Bi-MP$_{learned}$ | $s_{rel}$ |
|---|---|---|---|---|---|
| $\mathbf{A}_h$ $(10^1)$ | 1 | $1.517 \pm 0.054$ | $1.569 \pm 0.036$ | $1.585 \pm 0.029$ | |
| | 10 | $1.569 \pm 0.036$ | $1.607 \pm 0.021$ | $1.438 \pm 0.045$ | **0.133** |
| | 100 | $1.585 \pm 0.029$ | $1.651 \pm 0.053$ | $1.567 \pm 0.064$ | |
| Betweenness $(10^4)$ | 1 | $6.099 \pm 0.474$ | $5.965 \pm 0.401$ | $5.964 \pm 0.273$ | |
| | 10 | $5.909 \pm 0.261$ | $6.298 \pm 0.125$ | $6.418 \pm 0.275$ | **0.187** |
| | 100 | $6.018 \pm 0.257$ | $5.540 \pm 0.277$ | $6.692 \pm 0.420$ | |
| Closeness $(10^1)$ | 1 | $1.117 \pm 0.105$ | $0.938 \pm 0.061$ | $1.052 \pm 0.043$ | |
| | 10 | $1.028 \pm 0.054$ | $0.982 \pm 0.036$ | $1.036 \pm 0.051$ | **0.260** |
| | 100 | $1.152 \pm 0.123$ | $1.081 \pm 0.056$ | $1.201 \pm 0.059$ | |
| Eigenvector $(10^3)$ | 1 | $6.589 \pm 0.256$ | $6.580 \pm 0.088$ | $6.589 \pm 0.305$ | |
| | 10 | $6.608 \pm 0.114$ | $6.938 \pm 0.240$ | $6.468 \pm 0.117$ | **0.126** |
| | 100 | $6.668 \pm 0.099$ | $6.735 \pm 0.108$ | $6.835 \pm 0.076$ | |
| Degree $(10^0)$ | 1 | $40.017 \pm 3.009$ | $35.079 \pm 1.874$ | $38.011 \pm 1.209$ | |
| | 10 | $38.000 \pm 1.510$ | $36.868 \pm 0.871$ | $37.138 \pm 1.382$ | **0.419** |
| | 100 | $41.664 \pm 3.440$ | $39.831 \pm 1.498$ | $43.085 \pm 1.939$ | |
| Participation $(10^1)$ | 1 | $1.685 \pm 0.143$ | $1.533 \pm 0.123$ | $1.549 \pm 0.120$ | |
| | 10 | $1.613 \pm 1.067$ | $1.623 \pm 0.134$ | $1.471 \pm 0.166$ | **0.114** |
| | 100 | $1.646 \pm 0.162$ | $1.647 \pm 0.224$ | $1.709 \pm 0.063$ | |
| Clustering $(10^2)$ | 1 | $11.343 \pm 1.067$ | $9.844 \pm 0.603$ | $10.945 \pm 0.360$ | |
| | 10 | $10.436 \pm 0.694$ | $10.203 \pm 0.296$ | $10.458 \pm 0.846$ | **0.231** |
| | 100 | $10.978 \pm 0.645$ | $11.216 \pm 0.495$ | $11.952 \pm 0.591$ | |
| Small Worldness $(10^2)$ | 1 | $4.915 \pm 1.144$ | $4.265 \pm 1.009$ | $4.914 \pm 0.607$ | |
| | 10 | $4.200 \pm 0.696$ | $4.500 \pm 0.402$ | $5.129 \pm 0.704$ | **0.503** |
| | 100 | $5.608 \pm 1.370$ | $3.909 \pm 1.022$ | $5.275 \pm 0.569$ | |

