# OpenReview forum: "Rethinking Graph Super-Resolution: Dual Frameworks for Topological Fidelity"
_ICLR.cc/2025/Conference — ICLR 2025 Conference Withdrawn Submission_

### Official Review · Reviewer_cjZ7 · 2024-11-02

**Soundness:** 2
**Presentation:** 2
**Contribution:** 3
**Rating:** 5
**Confidence:** 2

**Summary:**

This paper investigates the graph super-resolution problem, aiming to construct a high-resolution graph from the low-resolution one. Two graph neural network (GNN)-based approaches termed Bi-SR and DEFEND are proposed in this paper, with theoretical analysis and discussions. Experiments are conducted to show the effectiveness  of the proposed methods.

**Strengths:**

1. The target research problem is rarely investigated by the community so far, and it is indeed a practical technique in several real-world applications, such as brain network analysis.
2. Ample discussion and theoretical analysis are provided to guarantee the effectiveness of the proposed method.
3. Experiments are conducted under diverse scenarios, from synthetic graphs to real-world brain graphs, which shows the usefulness of the proposed method.

**Weaknesses:**

1. The motivation and background of this paper is not clear. Specifically, in Line 064 the authors mentioned "the topological limitation". However, the specific definition of topological limitation is not clearly given in this part.
2. In the experiments, I found that there is few baseline for comparison. Is that because this research question is too new and hence no available methods for comparison? Do there exist more heuristic methods?
3. The presentation of this paper can be further improved. Specifically: 1. Notations in Line 102 Aij -> A_ij; 2. Figures 1 and 2 are not easy to read.

**Questions:**

1. Apart from the brain graphs, are there any practical applications of graph super-resolution?
2. How is the complexity of the proposed methods? Will they run slowly when the original low-resolution graph is already large, or when the super-resolution graph is large?

---

> ### Author Response · Authors · 2024-12-03
> **Response to Reviewer cjZ7**
>
> We thank the reviewer for their insightful comments and provide below responses to raised questions and identified weaknesses:
>
> > **W1: Clarify the use of the term “topological limitation”**
>
> The "topological limitation" mentioned in Line 064 refers to the limitations of existing graph super-resolution methods discussed in Lines 059–063. These methods primarily operate in the node space (i.e., learning signals on nodes) and have limited expressivity for modeling HR edges and graph topology. In the following paragraph, we explain the importance of accurately predicting graph topology during brain graph resolution, as it enhances neural fingerprinting and behavior prediction—both crucial for diagnosing neurodegenerative diseases such as Alzheimer’s and Parkinson’s.
>
> > **W2: Few comparison baselines**
>
> The reviewer is correct to infer that there are few baselines for comparison due to the nascency of this research direction and limited options available to the authors. We tried to overcome this issue by introducing additional baselines and datasets and conducting extensive ablation studies. While more thorough benchmarking is outside the scope of present work, we hope that it would encourage future work in this direction.
>
> > **Q1: Other practical applications of graph super-resolution**
>
> While our primary focus is on network neuroscience, graph super-resolution has potential applications in other domains, especially the -omics fields such as genomics and proteomics, where inferring high-resolution graphs from low-resolution data can reveal fine-grained structural and functional insights.
>
> > **Q2: Complexity for large graphs**
>
> The Bi-SR method has a quadratic complexity, which is comparable to applying standard message-passing operations (e.g., GCN, GAT, TransformerConv) on a dense graph. The DEFEND method has cubic complexity. Even with these complexities, our method is still computationally efficient compared to the SOTA, which faces out-of-memory (OOM) issues on dense brain graphs with >250 nodes. For reference, all experiments were performed on a single Tesla A100 10GB GPU. Moreover, even if this seems high for future applications on larger graphs, it is not necessarily a major disadvantage of the present work as such models regularly lead to impactful contributions in specific fields (see [1] which performs experiments only on the ZINC250k dataset with a maximum of 38 molecules).
>
> **References:**
>
> [1] Shi C, Xu M, Zhu Z, Zhang W, Zhang M, Tang J. GraphAF: a flow-based autoregressive model for molecular graph generation (2020). arXiv preprint arXiv:2001.09382. 2001 (ICLR 2020)

---

### Official Review · Reviewer_76pA · 2024-11-03

**Soundness:** 2
**Presentation:** 1
**Contribution:** 2
**Rating:** 3
**Confidence:** 4

**Summary:**

Graph super-resolution is an underexplored yet highly important research direction. Existing work on graph super-resolution leverages graph neural networks (GNNs), but they suffer from some limitations. In this paper, the authors propose two frameworks: Bi-SR, which performs structure-aware node super-resolution, and DEFEND, which focuses on edge representation learning for enhanced edge modeling. Experiments show the superiority of the proposed methods.

**Strengths:**

1. Graph super-resolution is an important problem, but underexplored in literature particularly in comparison to the image domain.
2. The proposed method shows a good empirical performance, specifically in the brain graph domain, where graph super-resolution is important.
3. The authors have well explored potential approaches to each sub-problem, not just presenting the final form of their methods, making this work beneficial to the community.

**Weaknesses:**

1. Writing should be definitely improved in general. (a) The organization of the paper is strange; it’s hard to separate previous works, their limitations, and what the authors propose throughout the paper. (b) Figures 1 and 2 are not mentioned in the text, and they can be located better. (c) I have other comments as well. Please see the questions below.
2. The technical novelty of the proposed methods is unclear. The methods consist of various different components, but I’m not sure what are their core improvements and what are technical details.
3. I believe that the scalability and speed can be an important issue of the proposed approach, but the experiments were done on small graph datasets. It would be nice to see experiments on large graphs.

**Questions:**

1. I think the authors can safely remove lines 138 - 143 from the paper.
2. Do all existing approaches for graph super-resolution have the limitation described in Section 2.4? Which of them have the limitation, and which do not?
3. Is it computationally feasible to create all-pair connections between the $n_l$ and $n_h$ nodes in Eq. (3)?
4. I’m not certain why we need the refinement step in Section 3.1, since the operation seems very similar to the super-resolution, which we aim to do eventually.
5. In Section 3.2, why do the authors use graph transformers instead of more lightweight and efficient graph neural networks?
6. How can we prove Proposition 1 and Corollary 1 if the terms like “edge space” and “more effective” are not properly defined?

---

> ### Author Response · Authors · 2024-12-03
> **Response to Reviewer 76pA**
>
> We thank the reviewers for their insightful comments and address raised questions and identified weaknesses below:
>
> > **W2: Clarification around technical novelty**
>
> In our work, we are proposing two operators:
>
> * **Bi-SR**: This operator focuses on increasing the number of fundamental building blocks (nodes) of the graph, analogous to how transposed convolutions increase the number of pixels in image super-resolution.
> * **DEFEND**: Once the HR graph structure is learned, this operator refines the signal (features) on the graph, akin to performing convolutions on the upscaled image.
>
> These components address limitations (absence of source node permutation invariance, mapping feature dimensions to nodes, and limited expressivity to model edge weights) of prior works. Moreover, they address complementary aspects of graph super-resolution. While Bi-SR is sufficient for learning both the data domain (graph) and the signal over it (node features), DEFEND provides an optional feature refinement step that we found to be particularly beneficial in certain applications, such as network neuroscience where we encounter dense brain graphs.
>
> > **W3: Scalability and speed**
>
> We appreciate the reviewer’s concern and add a detailed breakdown of time and memory complexity in the iterated version of the paper. While the worst-case complexity is quadratic for Bi-SR and cubic for DEFEND, our method is still computationally efficient compared to the SOTA, which faces out-of-memory (OOM) issues on dense brain graphs with >250 nodes. Even though the computational complexity may seem high for future applications on larger graphs, it is not necessarily a major disadvantage of the present work as such models regularly lead to impactful contributions in specific fields and have previously been accepted to ICLR (see [1] which performs experiments only on the ZINC250k dataset with a maximum of 38 molecules).
>
> > **Q2: Limitations of existing methods**
>
> All known graph super-resolution methods based on GNNs share this limitation. Specifically, [2, 3] employ the linear algebraic method discussed in the paper, while [4] utilizes eigenvalue decomposition and spectral feature mapping, introducing the same limitation in the spectral domain.
>
> > **Q3: Computational feasibility of bipartite connections**
>
> For our dense brain graphs, creating the n_l×n_h bipartite edges between LR and HR nodes is computationally feasible. The Bi-SR method scales quadratically with n_h​, which is comparable to applying standard message-passing operations (e.g., GCN, GAT, TransformerConv) on a dense graph. Note that our method significantly outperforms the current state-of-the-art, which encountered OOM errors and required adaptation for our dataset. For reference, all experiments were performed on a single Tesla A100 10GB GPU.
>
> > **Q4: Need for refinement step**
>
> The key distinction between the super-resolution and refinement steps lies in the underlying domains over which they perform message passing. The super-resolution step learns initial HR node features by only connecting LR and HR nodes. In contrast, the refinement step, which is optional, updates the learned HR features by incorporating messages from other HR nodes. This separation decouples two types of interactions: (1) between LR and HR nodes, which is essential, and (2) between HR nodes, which is optional but can provide empirical benefits.
>
> > **Q5: Use of graph transformers**
>
> In theory, our method can be combined with any message-passing layer. However, during experimentation we observed that simpler layers, such as GCN, perform poorly, possibly due to the dense nature of brain graphs and their tendency to oversmooth, even with a single GNN layer. TransformerConv effectively addresses this issue by selectively attending to relevant neighbors. Additionally, TransformerConv proved significantly more computationally efficient than NNConv layers, which are commonly used in existing graph super-resolution methods.
>
> **References:**
>
>  [1] Shi C, Xu M, Zhu Z, Zhang W, Zhang M, Tang J. GraphAF: a flow-based autoregressive model for molecular graph generation (2020). arXiv preprint arXiv:2001.09382. 2001 (ICLR 2020)
>
> [2] Islem Mhiri, Ahmed Nebli, Mohamed Ali Mahjoub, and Islem Rekik. Non-isomorphic intermodality graph alignment and synthesis for holistic brain mapping. In International Conference on Information Processing in Medical Imaging, pp. 203–215. Springer, 2021
>
> [3] Furkan Pala, Islem Mhiri, and Islem Rekik. Template-based inter-modality super-resolution of brain connectivity. In Predictive Intelligence in Medicine: 4th International Workshop, PRIME 2021, Held in Conjunction with MICCAI 2021, Strasbourg, France, October 1, 2021, Proceedings 4,pp. 70–82. Springer, 2021
>
> [4] Megi Isallari and Islem Rekik. Brain graph super-resolution using adversarial graph neural network with application to functional brain connectivity. Medical Image Analysis, 71:102084, 2021

---

### Official Review · Reviewer_k51c · 2024-11-04

**Soundness:** 1
**Presentation:** 2
**Contribution:** 2
**Rating:** 3
**Confidence:** 3

**Summary:**

This work focuses on generating high-resolution (HR) graphs from low-resolution (LR) graphs. The authors propose two methods, Bi-SR and DEFEND, both using Graph Neural Networks (GNNs) as the foundational structure. Bi-SR constructs a bipartite graph that connects nodes between the LR and HR graphs. It randomly initializes the HR node features, and then uses a message-passing on the bipartite graph to refine the HR node features. These features are subsequently refined through another intra-graph message-passing on the HR graph nodes. This refinement process uses either full-graph message passing on HR nodes or a learned structure specifically for HR message passing. The DEFEND model, on the other hand, focuses on edge representation learning by performing message passing on the line graph of the original graph, treating edges as nodes. The methods are evaluated on synthetic graphs and a brain graph dataset.

**Strengths:**

1. The paper includes numerous illustrations that clearly demonstrate how the method works and showcases the results.
2. The authors validate their approach on both synthetic and real-world graphs and the results seem to be promising.
3. In addition to the experiments, they also provide a theoretical analysis of their method's effectiveness.

**Weaknesses:**

1. **Clarity and Writing Issues**
   The paper has several writing issues that make it difficult to follow. For example, Section 2.4 starts with "However, this method disrupts structural integrity..." without clarifying what "this method" refers to. This paragraph closely resembles the third paragraph in the same subsection, possibly due to editing issues. Overall, the story and motivation in each section do not flow smoothly, making it hard to understand how different parts connect.

2. **Lack of Clear Motivation for High-Resolution Graphs**
   The motivation for using high-resolution graphs, especially for general graphs, is unclear. The reviewer, unfamiliar with brain activity analysis, found it challenging to grasp the value of high-resolution graphs for general applications. Without a compelling use case in brain graph datasets, the motivation for this approach seems weak for broader graph applications.

3. **Reinventing Existing Concepts**
   The paper introduces well-known concepts as if they are novel. For instance, the "dual graph" discussed here corresponds to the line graph, which has been widely used in previous GNN research [1, 2].

4. **Inaccurate Mathematical Language**
   Propositions and corollaries should use precise mathematical language. Phrases like "this method is more effective" lack rigor, as "more effective" is not a mathematically defined term, leading to an unclear use of formal notation.

5. **Limited Real-World Application**
   The paper only uses one real-world graph dataset, and the broader applications of this method remain unclear to the reviewer.

6. **High Computational Complexity**
   The method seems computationally expensive, requiring a full bipartite graph, a fully connected high-resolution graph, and a dual graph from the fully connected graph. However, the paper lacks an analysis of the time and memory complexity of this approach.

7. **Unclear Experimental Setup and Evaluation Metrics**
   The experimental setup and evaluation metrics are not well-defined. For instance, it's unclear how Mean Absolute Error (MAE) is measured when multiple valid permutations of the high-resolution graph could exist, making it a potential graph isomorphism problem. Calculating edge-wise MAE without addressing permutation invariance does not seem to be a valid evaluation approach.

8. **Non-standard Definition of MPNNs**
   Section 2.3 defines MPNNs in a non-standard way. The message function could be more complex, potentially including edge features and non-linear combinations of previous layer embeddings.

----------
 **Minor Points**:
- In Equation (4), the term $\mathbf{A}_f^{ref}$ is repeated twice.
- Line 152: "violets" should be corrected to "violates."

----------
[1] Cai, L., et al. "Line graph neural networks for link prediction." *IEEE Transactions on Pattern Analysis and Machine Intelligence* 44.9 (2021): 5103-5113.
[2] Chen, Z., Li, X., & Bruna, J. "Supervised community detection with line graph neural networks." *arXiv preprint arXiv:1705.08415* (2017).

**Questions:**

1. Could you explain more about the experimental setup and how your evaluation metric accounts for permutations of the nodes in the HR graph?
2. What are other potential applications of this method beyond brain analysis?
3. How can the HR graph be useful for downstream tasks? High-resolution images make sense for human visual systems, but graphs are not typically analyzed visually. For a computational system, theoretically, performing a task such as classification on the LR graph should be equivalent to first learning the HR graph and then performing the downstream task on it.

---

> ### Author Response · Authors · 2024-12-03
> **Response to Reviewer k51c (1/2)**
>
> We thank the reviewer for their comments and provide below clarification w.r.t questions and weaknesses:
>
> >**W2: Lack of Clear Motivation for High-Resolution Graphs**
>
> We appreciate the reviewer’s concerns and direct their attention to lines 047–053 and 064–072, where we elaborate on the utility of HR brain graphs. HR brain graphs improve neural fingerprinting and behavior prediction, which are vital for diagnosing neurodegenerative diseases such as Alzheimer’s and Parkinson’s. However, acquiring HR graphs is costly, making lightweight super-resolution methods extremely desirable. Additionally, neuroscientific analysis relies heavily on brain graph topology, making topology preservation a critical aspect of graph super-resolution methods. This discussion demonstrates how our work significantly contributes to brain graph super-resolution by ensuring topological fidelity.
>
> Moreover, we respectfully disagree with the reviewer’s assessment that our approach seems weak without the inclusion of broader graph applications. It is customary for ICLR to accept papers that make substantial contributions to specific domains, such as neuroscience (see [1,2]) or molecular generation (see [3]).
>
> > **W3: Reinventing Existing Concepts**
>
> We believe the reviewer may have misunderstood our claims. In fact, in Section 4 (line 353), we acknowledge that dual graphs (adjoint or line-(di) graphs) are established concepts from graph theory and do not intend to claim novelty in the concept itself. Our novel contribution lies in leveraging them for graph super-resolution. We also cited related work using line (or dual) graphs (line 090) to the best of our knowledge. We appreciate the reviewer pointing us to more such works, which we are happy to incorporate.
>
> > **W5 & Q2: Limited Real-World Application**
>
> While network neuroscience is our primary focus, we believe graph super-resolution has broad potential applications, especially in other omic fields such as genomics or proteomics, where it may be helpful to infer a larger graph from a smaller graph. Due to the nascency of graph SR, its utility in other domains remains underexplored. Nevertheless, as mentioned under response to W2, we do not believe it to be a limitation of present work as impactful papers in specialized fields are routinely accepted at ICLR ([1], [2]), even with experiments on a single-dataset ([3]). By designing additional synthetic experiments and positing our framework as a generic graph learning task, we hope to encourage further research beyond network neuroscience.
>
> > **W6: High Computational Complexity**
>
> We appreciate the reviewer’s concern. Appendix A.2 already includes computational analysis of the dual graph formulation and we will add a detailed breakdown of time and memory complexity in the iterated version of the paper. While the worst-case complexity is quadratic for Bi-SR and cubic for DEFEND, our method is still computationally efficient compared to the SOTA, which faces out-of-memory (OOM) issues on dense brain graphs with >250 nodes. Note that we ran all experiments on a single Tesla A100 10GB GPU. Also, even though the quadratic and cubic complexity may seem high for certain future applications, it is not necessarily a major disadvantage of the present work as such models regularly lead to impactful contributions in specific fields (see [3] which performs experiments only on the ZINC250k dataset with a maximum of 38 molecules).
>
>
> **References:**
>
> [1] Antoniades A, Yu Y, Canzano J, Wang W, Smith SL. Neuroformer: Multimodal and multitask generative pretraining for brain data. arXiv preprint arXiv:2311.00136. 2023 Oct 31 (ICLR 2024)
>
> [2] Karniol-Tambour O, Zoltowski DM, Diamanti EM, Pinto L, Brody CD, Tank DW, Pillow JW. Modeling state-dependent communication between brain regions with switching nonlinear dynamical systems. InThe Twelfth International Conference on Learning Representations (ICLR 2024)
>
> [3] Shi C, Xu M, Zhu Z, Zhang W, Zhang M, Tang J. GraphAF: a flow-based autoregressive model for molecular graph generation (2020). arXiv preprint arXiv:2001.09382. 2001 (ICLR 2020)

---

> > ### Author Response · Authors · 2024-12-03
> > **Response to Reviewer k51c (2/2)**
> >
> > > **W7 & Q1: Unclear Experimental Setup and Evaluation Metrics**
> >
> > We would like to direct the reviewer to Appendix E, which details our experimental setup and evaluation metrics. It also motivates our choice of topological metrics and highlights their utility in identifying key topological structures in network neuroscience.
> > Additionally, permutation invariance in evaluation metrics is only necessary when nodes are interchangeable. As discussed in Appendix C3, each node in the brain graph represents a specific brain region, making nodes unique and non-interchangeable. Therefore, including permutation-invariant metrics in this context would be undesirable. Note that this node uniqueness is already encoded in our HR graph generation process. In Bi-SR, we achieve this using unique positional encoding for HR nodes (lines 203–210), while in DEFEND, we use fixed HR node order and fixed mapping between primal edges and dual nodes (lines 375–376).
> >
> > Finally, we agree with the reviewer and recognize that permutation invariance of evaluation metrics may be required when extending graph super-resolution to other domains. However, developing such metrics lies beyond the scope of this work and should be left at the discretion of researchers adapting these techniques for specific use cases.
> >
> > > **W8: Non-standard Definition of MPNNs**
> >
> > While MPNNs can be arbitrarily complex, we chose this particular definition of MPNN to properly reflect and set the scope of our theoretical analysis in section A.3.2. This formulation encompasses most commonly used forms of message passing neural networks, including GCN, GIN, GAT, GraphTransformer, NNConv, and others. It also supports the inclusion of edge features and non-linear combinations through the parameter $\alpha_{ij}^{l}$, which can represent a constant, depend on individual nodes or their features (enabling non-linear aggregation and inclusion of edge features), or be stochastic (e.g. for neighborhood sampling).
> >
> > > **Q3: Utility of HR graphs for downstream tasks**
> >
> > While the proposition that "performing a task on the LR graph should theoretically be equivalent to first learning the HR graph and then performing the task on it" may hold in certain contexts, it is generally not true, particularly in the -omics field. For instance, in network neuroscience, structural and functional connectomes convey vastly different information, making classification on HR nodes distinct from classification on LR nodes.
> >
> > Additionally, many downstream tasks extend beyond simple classification. For example, analyzing clustering coefficients and small-world properties of nodes in HR brain graphs can help identify early-stage Parkinson’s disease patients [4]. Such tasks require the greater accuracy provided by HR graphs compared to LR graphs [5], underscoring the critical role of HR graphs in specific downstream applications.
> >
> > **References:**
> >
> > [4] Joana B Pereira, Dag Aarsland, Cedric E Ginestet, Alexander V Lebedev, Lars-Olof Wahlund, Andrew Simmons, Giovanni Volpe, and Eric Westman. Aberrant cerebral network topology and mild cognitive impairment in early parkinson’s disease. Human brain mapping, 36(8):2980–2995, 2015.
> >
> > [5] Ye Tian, BT Thomas Yeo, Vanessa Cropley, Andrew Zalesky, et al. High-resolution connectomic fingerprints: Mapping neural identity and behavior. NeuroImage, 229:117695, 2021.

---

### Official Review · Reviewer_hBND · 2024-11-04

**Soundness:** 2
**Presentation:** 1
**Contribution:** 2
**Rating:** 5
**Confidence:** 3

**Summary:**

This paper focuses on graph super-resolution, aiming at lifting the small graphs to large ones and preserving the distributions of graph structures and node features. The authors argue that previous methods ignore the underlying structures and fail to learn edge information. To address these issues, this paper proposes Bi-SR and DEFEND for node super-resolution and edge modeling, respectively. Experimental results demonstrate the effectiveness of the proposed method over baselines.

**Strengths:**

1. Graph super-resolution is a new task in the era of GNNs. It is claimed to be crucial for certain applications, such as neuroscience.

2. This paper identifies the weaknesses of previous methods and verifies them through extensive experiments.

**Weaknesses:**

1. There are already some GNNs that can be used for graph super-resolution, such as Graph u-nets (the authors also mentioned this paper). However, this paper does not take them into account. In contrast, this paper only uses the basic Linear Algebraic method and autoencoder as baselines, making the experiments unconvincing.

2. This paper proposes two methods, including Bi-SR and DEFEND. However, this relationship between them is not clear. Is there a principle on how to choose the algorithm? Should we use them together or should we use them separately?

3. Although graph super-resolution is a new task, the research on generating graphs with different sizes has been studied for a long time. The representative method is called graphon. Generally, graphon-based methods not only have good theoretical guarantees to preserve structural information but also model the edge distribution. In my view, current issues on graph super-resolution can be well addressed by the advanced graphon methods, such as IGNR [1].

4. The presentation of this paper needs improvement. The notation is confusing and the figures are not in pdf format.

[1] Implicit Graphon Neural Representation. AISTATS 2023.

**Questions:**

See weaknesses.

---

> ### Author Response · Authors · 2024-12-03
> **Response to Reviewer hBND**
>
> We thank the reviewer for their insightful comments and provide following clarifications to address identified technical weaknesses.
>
> > **W1: Lacks comparisons with established methods like Graph U-Net**
>
> Graph U-Net is an encoder-decoder architecture using skip connections to define a gUnpool operation to increase the size of the graph. From the graph U-Net paper [1]: *“we record the locations of nodes selected in the corresponding gPool layer and use this information to place nodes back to their original positions in the graph”*. This necessitates that the input and output graphs should be of the same size and thus, can not be used for graph super-resolution where the objective is to predict a larger graph from a smaller graph.
>
> Our objective to mention Graph U-Net in the introduction was to present the limitation of existing operators on graphs (gUnpool), highlight the structural differences between image and graph super-resolution, and to justify the need for our Bi-SR operator. Note that unlike image super-resolution, which leverages transposed convolutions due to the fixed underlying data domain (a grid), graph super-resolution requires simultaneously learning both the data domain (graph) and the signal (features).
>
> While we agree with the reviewer that more thorough comparisons are generally required, the nascency of graph super-resolution severely limits our options. We tried to overcome this issue by providing new baselines and datasets, comparing with current SOTA, and conducting extensive ablation studies. We hope that our work would encourage future research in this direction.
>
> [1] Gao H, Ji S. Graph u-nets. Ininternational conference on machine learning 2019 May 24 (pp. 2083-2092). PMLR
>
> > **W2: Relationship between Bi-SR and DEFEND is unclear**
>
> Bi-SR and DEFEND address complementary aspects of graph super-resolution:
>
> * Bi-SR: This operator focuses on increasing the number of fundamental building blocks (nodes) of the graph, analogous to how transposed convolutions increase the number of pixels in image super-resolution.
> * DEFEND: Once the HR graph structure is learned, this operator refines the signal (features) on the graph, akin to performing convolutions on the upscaled image.
>
> While Bi-SR is sufficient for learning both the data domain (graph) and the signal over it (node features), DEFEND provides an optional feature refinement step that we found to be particularly beneficial in certain applications, such as network neuroscience where we encounter dense brain graphs.
>
> In principle, Bi-SR and DEFEND can be used independently or together to perform graph super-resolution. However, we would like to point out that even though DEFEND without Bi-SR may achieve competitive performance, it loses structural properties (such as source node invariance) that may be essential to certain tasks.
>
> > **W3: Advanced graphon-based methods like IGNR are not considered**
>
> We thank the reviewer for pointing us to this work. From our understanding of the paper, while IGNR can learn to generate graphs of different sizes, they are most effective when LR and HR graphs are derived from the same implicit representation. However, we frequently encounter the issue of distribution shift in graph super-resolution i.e. the LR and HR graphs may have different implicit representations. This is particularly common in network neuroscience (brain connectomics) and other omics fields. For example, structural and functional connectomes differ substantially in their underlying graph distributions [2]. This distributional shift may pose challenges for methods like IGNR. Having said so, we agree that IGNR could form a valid comparison baseline. However, adapting it to graph super-resolution involves additional training challenges that are beyond the scope of this work:
> * **Should we use  both LR-HR graphs during training?** This approach struggles due to distribution shifts.
> * **Should we perform conditional generation of HR graphs using LR graphs?** This requires encoding the LR graph as a vector which may be problematic due to lossy compression of information and potential loss of structural and discriminative features.
>
> In contrast, our Bi-SR operator directly learns the mapping between LR and HR graph domains, providing the flexibility needed to handle these distributional differences.
>
> [2] Park HJ, Friston K. Structural and functional brain networks: from connections to cognition. Science. 2013 Nov 1;342(6158):1238411

---

### Note · Authors · 2024-12-03

**Comment:**

We sincerely thank all the reviewers for their time, thoughtful comments, and insightful questions. Your feedback has been invaluable in helping us identify areas where the paper’s presentation, motivation, and writing can be improved.

While we have made a sincere effort to address the technical concerns raised, we recognize that further improvements are needed to enhance the clarity and communication of our work. To address these comprehensively, we have decided to withdraw the current submission and focus on creating an improved version of the paper.

Even as we withdraw, we are happy to engage in further discussions should there be additional feedback or suggestions from your end. We greatly appreciate your constructive input, which will undoubtedly strengthen our work and contribute to this exciting new direction.

**Withdrawal Confirmation:**

I have read and agree with the venue's withdrawal policy on behalf of myself and my co-authors.